# Importance Weighting for Aligning Language Models under Deployment Distribution Shift

**Thanawat Lodkaew**                                              *lodkaew@ms.k.u-tokyo.ac.jp*
*The University of Tokyo, Japan*

**Tongtong Fang**                                                        *fang@ism.ac.jp*
*The Institute of Statistical Mathematics, Japan*

**Takashi Ishida**                                                      *ishi@k.u-tokyo.ac.jp*
*RIKEN, Japan*
*The University of Tokyo, Japan*

**Masashi Sugiyama**                                                   *sugi@k.u-tokyo.ac.jp*
*RIKEN, Japan*
*The University of Tokyo, Japan*

**Reviewed on OpenReview:** *https://openreview.net/forum?id=C7QWN4AXvp*

## Abstract

Aligning language models (LMs) with human preferences remains challenging partly because popular approaches, such as reinforcement learning from human feedback and direct preference optimization (DPO), often assume that the training data is sufficiently representative of the environment in which the model will be deployed. However, real-world applications frequently involve distribution shifts, e.g., changes in end-user behavior or preferences during usage or deployment, which pose a significant challenge to LM alignment approaches. In this paper, we propose an importance weighting method tailored for DPO, namely IW-DPO, to address distribution shifts in LM alignment. IW-DPO can be applied to joint distribution shifts in the prompts, responses, and preference labels without explicitly assuming the type of distribution shift. Our experimental results on various distribution shift scenarios demonstrate the usefulness of IW-DPO.

## 1 Introduction

While language models (LMs) have been rapidly increasing their language generation capabilities in recent years, aligning them with human values and norms remains a challenging task (Shen et al., 2023). Among the various approaches for alignment, reinforcement learning from human feedback (RLHF) has demonstrated considerable success in aligning LMs with human preferences (Ziegler et al., 2019; Stiennon et al., 2020; Ouyang et al., 2022). However, it is involved in a rather complex training pipeline: reward modeling (RM) from preference data and optimization of an LM using a learned reward model and a reinforcement learning (RL) algorithm. To reduce this complexity, Rafailov et al. (2024) developed a simple yet effective optimization approach, namely direct preference optimization (DPO). DPO directly optimizes the LM without the need for RM and RL, thus making it simpler and faster.

DPO has been demonstrated to be an effective method for fine-tuning LMs to generate responses that align with human-desired outputs, leading to the creation of several widely used foundation LM families, such as Llama 3 (Grattafiori et al., 2024), Phi-4 (Abdin et al., 2024), Qwen2 (Yang et al., 2024), and DeepSeek (Bi et al., 2024). Like other machine learning algorithms (Quiñonero-Candela et al., 2008; Pan & Yang, 2009; Sugiyama & Kawanabe, 2012), however, DPO typically suffers from various distribution shifts that present a

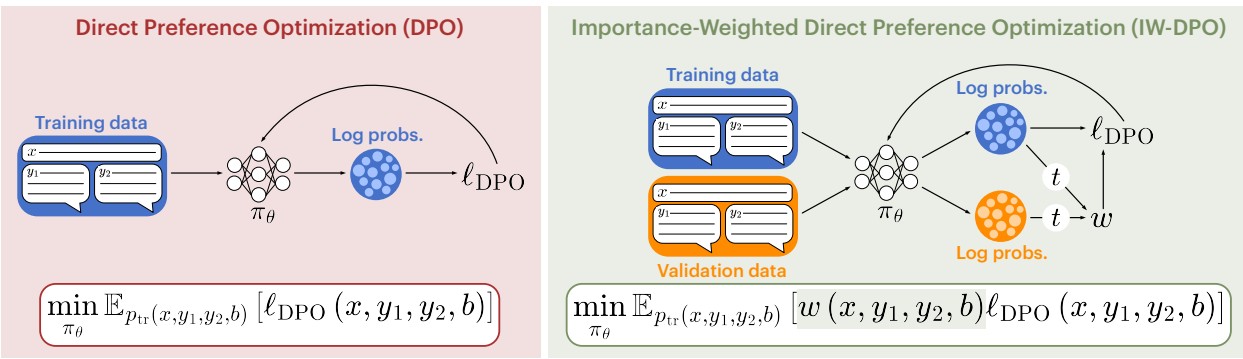

Figure 1: DPO optimizes for the training distribution by using only the training data, while **IW-DPO optimizes for the test distribution by additionally using a tiny amount of data (i.e., validation data) sampled from the test distribution to estimate weights and reweight training losses.** In weight estimation, the log probabilities of the training data and those of the validation data are passed through a transformation function $t$, and the transformed data are then used to compute the importance weights.

challenge in aligning with human-desired responses, underscoring the need for the development of a method that can effectively address such practical difficulties.

Recent studies have attempted to address the issue of distribution shifts in DPO, where the LM being optimized gradually deviates from the initial reference model (the LM used as initial weights for training) as training progresses on a fixed offline preference dataset, which we refer to as *model distribution shift*. For instance, Sun et al. (2023) proposed a way to address the difference between the reward distribution of the LM and that of the reference model. Gou & Nguyen (2024), Zhou et al. (2024) and Xu et al. (2024) explored a phenomenon in which the output (also called sample, response or completion in various literature) distribution of the LM changes, causing it to diverge from the distribution present in the fixed offline preference dataset. Similarly, Dou et al. (2024) examined how output distribution shifts negatively impact the performance of the reward model, diminishing its ability to distinguish between responses.

In contrast, our work addresses a fundamentally different form of distribution shift which we call the *deployment distribution shift*, where the environment changes in ways not reflected in the training dataset. Such shifts can arise from real-world usage or deployment, such as changes in end-user behavior or preferences. For the remainder of this paper, we will use the term "distribution shift" to denote this phenomenon. We characterize the factors that cause distribution shifts in LM alignment and, accordingly, systematically define the types of distribution shifts. Specifically, in the context of LM alignment, preference data typically consists of three elements: prompt, response, and preference label. Various types of distribution shifts between the training distribution and post-deployment, i.e., the test distribution, can arise from one or more of these factors. When a distribution shift occurs, training on the training dataset means optimizing for the training distribution, which may result in poor performance on the test distribution. Son et al. (2025) explored a shift in one of these factors, the preference shift problem, but focused on an online setting, whereas we assume to have a fixed offline preference dataset for training. We provide a detailed explanation of the definition of distribution shift, the contributing factors, and the types of distribution shifts in Section 3.1.

For solving distribution shift problems, importance weighting is a powerful tool that estimates a test-over-training density ratio as weights and uses these weights to reweight the training losses (Sugiyama & Kawanabe, 2012). Later, dynamic importance weighting (DIW) was proposed as a modern implementation of importance weighting, which makes it well suited for deep learning (Fang et al., 2020). However, DIW mainly focuses on classification, and its effectiveness in large-scale machine learning problems such as LM alignment has yet to be investigated.

In this paper, inspired by DIW, we propose an importance-weighted DPO, namely IW-DPO, to solve the distribution shifts in LM alignment. An overview of our method compared to the original DPO is shown

in Figure 1. IW-DPO estimates importance weights for training instances and uses them to up/down-weight training instances that are relevant/irrelevant to ensure that the LM is not overfitted to the training distribution and more aligned with the test distribution. To estimate the importance weights, IW-DPO uses a transformation function $t$ derived from the LM to convert raw preference data into low-dimensional representations. It then performs existing density ratio estimation methods, such as kernel mean matching (KMM) (Huang et al., 2006), Kullback–Leibler importance estimation procedure (KLIEP) (Sugiyama et al., 2007) and relative unconstrained least-squares importance fitting (RuLSIF) (Yamada et al., 2011), on the transformed data.

A significant advantage of IW-DPO is its capability to handle joint distribution shifts without requiring prior knowledge of the types of distribution shifts involved, making it particularly valuable for practical applications. To evaluate its effectiveness, we design and conduct experiments under various distribution shift scenarios in LM alignment. The results show a great potential of IW-DPO in handling practical distribution shift problems.

## 2 Related Work and Background

In this section, we first explore various approaches to importance weighting in LMs and then provide the background information on reward-based and reward-free RLHF. See Appendix D for a further discussion of related work.

### 2.1 Importance Weighting in LMs

Several approaches based on importance weighting have been proposed for language modeling. Grangier et al. (2023) proposed an importance weighting method for LM pre-training and fine-tuning, where importance weights are estimated by a separate weighting model trained jointly with the LM. While they concentrated on LM pre-training and fine-tuning, our emphasis is on LM alignment, particularly preference optimization. Moreover, our IW-DPO utilizes a transformation function from the LM for weight estimation, eliminating the need for joint training of a weighting model as they did. Jiang et al. (2024) applied importance weighting as a form of importance sampling to filter out self-generated examples that deviate from the desired distribution, aiming for self-improvement in LMs. However, their approach is constrained to datasets or tasks with clear, definitive answers because it includes components such as self-consistency and majority voting, whereas our focus is on more open-ended tasks. Zhou et al. (2024) proposed an extension of DPO, namely weighted preference optimization (WPO). Their approach involves reweighting training instances to address the distribution shift between the output distribution of the LM and the distribution presented in the training preference dataset. While WPO focuses on the model distribution shift, we focus on the deployment distribution shift. In addition, while WPO uses length-normalized sequence probabilities (i.e., probabilities of all predicted tokens in the response) as weights, IW-DPO estimates weights by using a density ratio estimation method. Sow et al. (2025) introduced an importance weighting method for LM pre-training with weight estimation based on training losses; however, they did not account for any distribution shifts. Additionally, their weight estimation method computes importance weights solely based on information from the training examples, specifically using training loss values. In contrast, our weight estimation process accounts for using information from both training and test distributions, thereby optimizing specifically for the test distribution.

### 2.2 RLHF

**Reward-based RLHF** In reward-based RLHF, following the pipeline in Stiennon et al. (2020), we first construct a reward model that approximates human preferences based on a pair of responses $(y_1, y_2)$ to a given prompt $x$.[1] Human annotators express a preference for one response over the other, referred to as preference label $b$, which is used to train the reward model. We define $b = +1$ if $y_1$ is preferred, and $b = -1$ if $y_2$ is preferred. One common approach for modeling human preferences is the Bradley-Terry model (Bradley

---

[1]Some RLHF pipelines, such as those in Ziegler et al. (2019) and Ouyang et al. (2022), may utilize more than two responses.

& Terry, 1952), which defines the preference probability expressed as

$$p(b \mid x, y_1, y_2) = \sigma \left( b \cdot (r^*(x, y_1) - r^*(x, y_2)) \right), \tag{1}$$

where $r^*$ is a latent reward model and $\sigma(u) = \frac{1}{1+\exp(-u)}$ is the sigmoid function. We are given a preference dataset $\mathcal{D} = \{(x^i, y_1^i, y_2^i, b^i)\}_{i=1}^N$ of $N$ instances. During the RM phase, we aim to optimize the following objective to train a reward model $r_\psi$ parameterized by $\psi$:

$$\min_{r_\psi} \mathbb{E}_{(x, y_1, y_2, b) \sim \mathcal{D}} \left[ -\log \sigma \left( b \cdot (r_\psi(x, y_1) - r_\psi(x, y_2)) \right) \right]. \tag{2}$$

After training the reward model, we proceed to the RL phase where we consider optimizing an LM $\pi_\theta$ parameterized by $\theta$.[2] The goal of this phase is to maximize the expected reward assigned to the generated response of the LM $\pi_\theta$ while ensuring that it does not drift too far from the reference model $\pi_{\text{ref}}$. This can be done by utilizing proximal policy optimization (Schulman et al., 2017), which results in the following objective:

$$\max_{\pi_\theta} \mathbb{E}_{x \sim \mathcal{D}_x, y \sim \pi_\theta(y|x)} \left[ r_\psi(x, y) \right] - \beta \mathbb{D}_{\text{KL}} \left[ \pi_\theta(\cdot \mid x) \| \pi_{\text{ref}}(\cdot \mid x) \right], \tag{3}$$

where $y$ is a response generated by $\pi_\theta$ given a prompt $x$ sampled from $\mathcal{D}_x = \{x^i\}_{i=1}^N$, and $\mathbb{D}_{\text{KL}}$ is the Kullback–Leibler (KL) divergence and ensures that the LM does not diverge too far from the reference model, as controlled by a hyperparameter $\beta > 0$.

**Reward-free RLHF**   DPO (Rafailov et al., 2024) simplifies the RLHF process by directly optimizing the LM using human preference data, without the need for RM and RL. The derivation of the DPO loss begins by reparameterizing the reward function in terms of the LM $\pi_\theta$ and the reference model $\pi_{\text{ref}}$, resulting in an implicit reward function $r$. We can then express the probability of human preferences in terms of the LM directly, thereby bypassing the need to fit an explicit reward model (Rafailov et al., 2024). This results in the DPO loss, which is defined as

$$\ell_{\text{DPO}}(x, y_1, y_2, b) = -\log \sigma \left( b \cdot (r(x, y_1) - r(x, y_2)) \right), \tag{4}$$

where the implicit reward function $r$ is given by

$$r(x, y) = \beta \log \frac{\pi_\theta(y \mid x)}{\pi_{\text{ref}}(y \mid x)}. \tag{5}$$

Going forward, we will omit $x$ in Eq. (5) for simplicity. In practice, a simple way to derive $\pi_\theta$ and $\pi_{\text{ref}}$ is to initialize them to a supervised fine-tuned LM (Rafailov et al., 2024), which we will refer to as supervised fine-tuning (SFT).[3]

## 3   Proposed Method

In this section, we introduce the mechanism of IW-DPO. We begin by providing an explanation of the definition of distribution shift and formulating the objective that we aim to optimize. Next, we describe how to optimize this objective using IW-DPO. Finally, we present two variants of IW-DPO.

### 3.1   Problem Setting

**Distribution shift**   A shift in the distribution of the data is defined as the underlying joint density of the training preference data $p_{\text{tr}}(x, y_1, y_2, b)$ differing from that of the test preference data $p_{\text{te}}(x, y_1, y_2, b)$, i.e., $p_{\text{tr}}(x, y_1, y_2, b) \neq p_{\text{te}}(x, y_1, y_2, b)$.

---

[2]Given a prompt $x$, $\pi_\theta$ generates a response $y$ in an auto-regressive manner characterized by $\pi_\theta(y \mid x) = \prod_j \pi_\theta \left( y_j \mid x, y_{<j} \right)$, where $y_j$ is the $j$-th token in the response and $y_{<j}$ is the tokens in the response prior to $y_j$ (Xu et al., 2024).

[3]In RLHF, SFT typically involves fine-tuning an LM on pairs of prompts and their corresponding responses (Ouyang et al., 2022).

**Factors of distribution shift** The factors contributing to distribution shift can be categorized by expressing the joint density as $p(x, y_1, y_2, b) = p(x)p(y_1, y_2 \mid x)p(b \mid x, y_1, y_2)$ and studying each component individually: 1) **Prompt:** A change in prompts may arise from a shift in the domain of interest, such as from culinary topics to agricultural practices. Formally, this sort of change can be expressed as $p_{\text{tr}}(x) \neq p_{\text{te}}(x)$. 2) **Response:** A change in responses may result from a shift in the preferred response style. For instance, whereas helpful responses were previously expected, there is now an expectation for responses to be both helpful and harmless. Formally, this change in responses can be expressed as $p_{\text{tr}}(y_1, y_2 \mid x) \neq p_{\text{te}}(y_1, y_2 \mid x)$. 3) **Preference label:** A change in user preferences can lead to a shift in the distribution of preference labels, even if the prompts and responses remain unchanged. Formally, this change in preference labels can be expressed as $p_{\text{tr}}(b \mid x, y_1, y_2) \neq p_{\text{te}}(b \mid x, y_1, y_2)$.

Table 1: Factors and potential distribution shift types. Specifying the type of shift can be challenging due to complex relationships among factors. Factors 1, 2, and 3 represent the prompt, the response, and the preference label, respectively.

| Type of shift | Factor | | |
|---|---|---|---|
| | 1 | 2 | 3 |
| a   No shift | | | |
| b   Full shift | ✓ | ✓ | ✓ |
| c   Prompt shift | ✓ | | |
| d   Response shift | | ✓ | |
| e   Preference label shift | | | ✓ |
| f   Prompt + response shift | ✓ | ✓ | |
| g   Prompt + preference label shift | ✓ | | ✓ |
| h   Response + preference label shift | | ✓ | ✓ |

A distribution shift can be caused by one or more of these factors, resulting in different types of distribution shifts. In this paper, we consider the full distribution shift, which includes all seven specific types as special cases. We show the relationship between the causes and all possible distribution shift types in Table 1. Although there are multiple factors and distribution shift types, we will demonstrate that our method can effectively address such distribution shift problems without requiring knowledge of the underlying factors or specific types of distribution shifts.

**Learning objective** Ideally, the LM $\pi_\theta$ should be learned by optimizing the following objective:

$$\mathcal{J}(\pi_\theta) = \mathbb{E}_{p_{\text{te}}(x, y_1, y_2, b)}\left[\ell_{\text{DPO}}(x, y_1, y_2, b)\right]. \tag{6}$$

When the training and test distributions differ, training solely on the training data implies that we are optimizing for the training distribution, which may lead to suboptimal performance on the test distribution. In addition to a training preference dataset from the training distribution $\mathcal{D}_{\text{tr}} = \{(x^{\text{tr},i}, y_1^{\text{tr},i}, y_2^{\text{tr},i}, b^{\text{tr},i})\}_{i=1}^{N_{\text{tr}}} \overset{\text{i.i.d.}}{\sim} p_{\text{tr}}(x, y_1, y_2, b)$, our problem setting further assumes the availability of a validation preference dataset from the test distribution $\mathcal{D}_{\text{v}} = \{(x^{\text{v},i}, y_1^{\text{v},i}, y_2^{\text{v},i}, b^{\text{v},i})\}_{i=1}^{N_{\text{v}}} \overset{\text{i.i.d.}}{\sim} p_{\text{te}}(x, y_1, y_2, b)$. However, the size of $\mathcal{D}_{\text{v}}$ is considerably smaller than that of $\mathcal{D}_{\text{tr}}$, i.e., $N_{\text{v}} \ll N_{\text{tr}}$. This reflects a real-world situation in which it may be possible to collect a limited amount of preference data from the test distribution. We can use $\mathcal{D}_{\text{v}}$ to directly approximate Eq. (6), but it may not be accurate due to the limited sample. Hence, our goal is to utilize both $\mathcal{D}_{\text{tr}}$ and $\mathcal{D}_{\text{v}}$ to learn $\pi_\theta$ that makes Eq. (6) small. Given that the size of $\mathcal{D}_{\text{v}}$ is tiny, we anticipate that utilizing $\mathcal{D}_{\text{v}}$ during training with $\mathcal{D}_{\text{tr}}$ will yield better performance than training with $\mathcal{D}_{\text{v}}$ alone (Fang et al., 2020).

## 3.2 Importance-Weighted DPO

In this section, we first present the derivation of the training objective and then describe the procedure for weight estimation.

### 3.2.1 Training Objective

To make the learning objective in Eq. (6) small, we propose an importance-weighted DPO method, which we call IW-DPO. Let supp$(p)$ denote the *support* of a density $p$ over $(x, y_1, y_2, b)$, defined as the set of points where $p$ assigns nonzero probability density. As is standard in importance weighting, we assume that the

support of the training distribution covers that of the test distribution; that is, $\text{supp}(p_{\text{te}}) \subseteq \text{supp}(p_{\text{tr}})$. Under this assumption, the importance weight $w^*(x, y_1, y_2, b) = p_{\text{te}}(x, y_1, y_2, b)/p_{\text{tr}}(x, y_1, y_2, b)$ is well-defined, and the following proposition holds.

**Proposition 1** *Given the true importance weight $w^*$ and $\pi_\theta$ that minimize the importance-weighted risk on the training distribution $\mathcal{J}_{\text{tr}}(\pi_\theta, w^*)$, the risk on the test distribution (Eq. (6)) can be expressed as*

$$\mathcal{J}(\pi_\theta) = \mathbb{E}_{p_{\text{tr}}(x, y_1, y_2, b)}\left[w^*(x, y_1, y_2, b)\ell_{\text{DPO}}(x, y_1, y_2, b)\right] = \mathcal{J}_{\text{tr}}(\pi_\theta, w^*).$$

The proof is shown in Appendix A. Proposition 1 implies that minimizing the importance-weighted risk on the training distribution is equivalent to minimizing the risk on the test distribution. In practice, it is necessary to estimate $w^*$ since it is unknown. $\mathcal{J}_{\text{tr}}$ can be approximated by the weighted empirical loss over the training distribution. Formally, an importance-weighted empirical version of $\mathcal{J}$ (as defined in Eq. (6)) is given by

$$\hat{\mathcal{J}}(\pi_\theta) = \frac{1}{N_{\text{tr}}} \sum_{i=1}^{N_{\text{tr}}} w^{\text{tr},i} \ell_{\text{DPO}}(x^{\text{tr},i}, y_1^{\text{tr},i}, y_2^{\text{tr},i}, b^{\text{tr},i}), \tag{7}$$

where $w^{\text{tr},i}$ is the empirical importance weight of the $i$-th training instance. If the empirical weights $w^{\text{tr},i}$ equal the true importance weights $w^*$, then—by Proposition 1—$\hat{\mathcal{J}}$ is an unbiased estimator of $\mathcal{J}$. More generally, when $w^{\text{tr},i}$ are consistent estimates of $w^*$, $\hat{\mathcal{J}}$ remains a consistent (asymptotically unbiased) estimator.

**Why is importance weighting important?** As discussed in Proposition 1, when considering $w$, we can ensure that minimizing the risk on the training distribution aligns with minimizing the risk on the test distribution, effectively optimizing $\pi_\theta$ for the test distribution. However, in the absence of $w$ (i.e., when $\min_{\pi_\theta} \mathbb{E}_{p_{\text{tr}}(x, y_1, y_2, b)}\left[\ell_{\text{DPO}}(x, y_1, y_2, b)\right]$), there is no guarantee that minimizing the risk on the training distribution will correspond to minimizing the risk on the test distribution. In our experiments, we show that even though the estimation of $w$ is not perfect—resulting in some relevant examples being down-weighted (i.e., $w \ll 1$) and some irrelevant examples being up-weighted (i.e., $w \gg 1$)—our methods still consistently outperform the baseline methods. For more details, see Section 4.2.1, particularly Figure 2.

### 3.2.2 Weight Estimation

The key challenge then becomes how to estimate the importance weights. Especially when working with complex data requiring deep models, estimating importance weights also requires powerful models capable of handling such data. One straightforward approach is to model the importance weights $w^*(x, y_1, y_2, b)$ directly with a deep neural network, which requires joint training of both an LM and a separate weighting model (Grangier et al., 2023). In contrast, we adopt a simpler approach that uses a transformation function derived from the LM to transform the inputs into the low-dimensional transformed data (Fang et al., 2020).

In particular, we introduce a transformation function $t : (x, y_1, y_2, b) \mapsto z$ that transforms $\mathcal{D}_{\text{tr}}$ and $\mathcal{D}_{\text{v}}$ into a set of transformed training data $\mathcal{Z}_{\text{tr}} = \{z^{\text{tr},1}, \ldots, z^{\text{tr},N_{\text{tr}}}\}$ and transformed validation data $\mathcal{Z}_{\text{v}} = \{z^{\text{v},1}, \ldots, z^{\text{v},N_{\text{v}}}\}$. We then estimate importance weights by applying a density ratio estimation method to the transformed data $\mathcal{Z}_{\text{tr}}$ and $\mathcal{Z}_{\text{v}}$. Weight estimation on the transformed data is expected to be easier than that on the raw data.

While several density ratio estimation methods can be applied, we use KMM (Huang et al., 2006) to illustrate how to derive the importance weights for clarity and brevity. In our experiments, we also explore alternative methods, including KLIEP (Sugiyama et al., 2007) and RuLSIF (Yamada et al., 2011), and discuss them in Section 4.2.4.

In KMM, our objective is to determine importance weights $w^{\text{tr},1}, \ldots, w^{\text{tr},N_{\text{tr}}}$ that ensure the mean embedding of the training distribution is approximately equal to that of the test distribution within a reproducing kernel Hilbert space $\mathcal{H}$. It is known that there exists a feature map $\phi : \mathbb{R}^d \to \mathcal{H}$ such that $k(z, z') = \langle\phi(z), \phi(z')\rangle_{\mathcal{H}}$, where $d$ is the dimension of the transformed data $z$ and $\langle\cdot, \cdot\rangle_{\mathcal{H}}$ represents the inner product in $\mathcal{H}$ (Smola et al., 2007). Then, let $\mu_{\text{tr}} = \mathbb{E}_{p_{\text{tr}}(x, y_1, y_2, b) \cdot w(z)}[\phi(z)]$ and $\mu_{\text{te}} = \mathbb{E}_{p_{\text{te}}(x, y_1, y_2, b)}[\phi(z)]$ represent the kernel embeddings

of $p_{\text{tr}}(x, y_1, y_2, b) \cdot w(z)$ and $p_{\text{te}}(x, y_1, y_2, b)$ in $\mathcal{H}$, respectively. Subsequently, KMM aims to minimize the discrepancy between $\mu_{\text{tr}}$ and $\mu_{\text{te}}$, which can be estimated with two empirical means as follows:

$$\|\mu_{\text{tr}} - \mu_{\text{te}}\|_{\mathcal{H}}^2 \approx \left\| \frac{1}{N_{\text{tr}}} \sum_{i=1}^{N_{\text{tr}}} w^{\text{tr},i} \phi\left(z^{\text{tr},i}\right) - \frac{1}{N_{\text{v}}} \sum_{i=1}^{N_{\text{v}}} \phi\left(z^{\text{v},i}\right) \right\|_{\mathcal{H}}^2 = \frac{1}{N_{\text{tr}}^2} \boldsymbol{w}^\top \boldsymbol{K} \boldsymbol{w} - \frac{2}{N_{\text{tr}}^2} \boldsymbol{k}^\top \boldsymbol{w} + \text{const.}, \qquad (8)$$

where $\boldsymbol{w} = [w^{\text{tr},1}, \ldots, w^{\text{tr},N_{\text{tr}}}]$ is the weight vector, $\boldsymbol{k}_i = \frac{N_{\text{tr}}}{N_{\text{v}}} \sum_{j=1}^{N_{\text{v}}} k\left(z^{\text{tr},i}, z^{\text{v},j}\right)$, $\boldsymbol{K}_{ij} = k(z^{\text{tr},i}, z^{\text{tr},j})$, and "const." is a constant that does not depend on $\boldsymbol{w}$. As a kernel function, we use the radial basis function (RBF) (Buhmann, 2000) in this work, i.e., $k(z, z') = \exp\left(-\gamma \|z - z'\|^2\right)$, where $\gamma > 0$ is the kernel width parameter. More formally, KMM solves the following quadratic optimization problem for $\boldsymbol{w}$:

$$\min_{\boldsymbol{w}} \frac{1}{2} \boldsymbol{w}^\top \boldsymbol{K} \boldsymbol{w} - \boldsymbol{k}^\top \boldsymbol{w} + \lambda \|\boldsymbol{w}\|_2^2 \ \text{ subject to } w^{\text{tr},i} \in [0, B] \text{ and } \left| \frac{1}{N_{\text{tr}}} \sum_{i=1}^{N_{\text{tr}}} w^{\text{tr},i} - 1 \right| \leq \epsilon, \qquad (9)$$

where $\lambda > 0$ is the $\ell_2$ regularization hyperparameter, $B > 0$ is an upper bound of the weights, and $\epsilon > 0$ is a slack variable.

As the use of $(x, y_1, y_2, b)$ is not straightforward for density ratio estimation, it is necessary to properly define the transformation function $t$. In Section 3.3, we will explain different choices of $t$.

### 3.3 Choices of Transformation Function

Here we explain how we can use $\ell_{\text{DPO}}$ (Eq. (4)) and $r$ (Eq. (5)) as $t$.

#### 3.3.1 Loss

Fang et al. (2020) suggested using *loss* values (i.e., using $\ell_{\text{DPO}}$ (Eq. (4))) to estimate importance weights, which in our case corresponds to $t : (x, y_1, y_2, b) \mapsto \ell_{\text{DPO}}(x, y_1, y_2, b)$. We denote this method as IW-DPO-Loss or IW-DPO-L for short.

**Issue of IW-DPO-L**  Using loss values for weight estimation can be problematic, because $\ell_{\text{DPO}}$ is not invertible. For example, a loss value can be associated with multiple instances $(x, y_1, y_2, b)$ as long as their reward margins (i.e., $r(y_1) - r(y_2)$) are identical. As stated in Fang et al. (2020), $t$ cannot be arbitrarily defined but it must ideally satisfy three properties: fixed, deterministic, and invertible. Although $\ell_{\text{DPO}}$ is fixed and deterministic, it is not invertible, and thus technically not a proper transformation function.

#### 3.3.2 Reward

To avoid the issue of IW-DPO-L, we propose utilizing *reward* values (i.e., utilizing $r$ (Eq. (5))) in place of loss values as transformed data during weight estimation. Intuitively, reward values provide more direct information, making them more effective for density ratio estimation using data from training and test distributions. Since we have two responses $(y_1, y_2)$ for each prompt $x$, we suggest using the reward values of both responses because we may lose information if we use only one of them. Formally, we have $t : (x, y_1, y_2, b) \mapsto \hat{r}(x, y_1, y_2, b)$, where $\hat{r}(x, y_1, y_2, b) = (r(y_1), r(y_2))$ is a tuple-valued function. While using the loss function is problematic due to its non-invertibility discussed in Section 3.3.1, we use the reward values to avoid the issue.

**Kernel combination**  Given that $\hat{r}$ does not output a scalar but a tuple of two reward values, we have $\mathcal{Z}_{\text{tr}} = \{(z_{y_1}^{\text{tr},1}, z_{y_2}^{\text{tr},1}), \ldots, (z_{y_1}^{\text{tr},N_{\text{tr}}}, z_{y_2}^{\text{tr},N_{\text{tr}}})\}$ and $\mathcal{Z}_{\text{v}} = \{(z_{y_1}^{\text{v},1}, z_{y_2}^{\text{v},1}), \ldots, (z_{y_1}^{\text{v},N_{\text{v}}}, z_{y_2}^{\text{v},N_{\text{v}}})\}$, where $z_{y_1}$ and $z_{y_2}$ correspond to $r(y_1)$ and $r(y_2)$, respectively, and we cannot compute $k$ directly. To address this, we compute two kernels for $z_{y_1}$ and $z_{y_2}$ separately and combine them. Specifically, we combine the two kernels by multiplying them together. Then, in Eq. (9), we have $\boldsymbol{k}_i = \frac{N_{\text{tr}}}{N_{\text{v}}} \sum_{j=1}^{N_{\text{v}}} k(z_{y_1}^{\text{tr},i}, z_{y_1}^{\text{v},j}) k(z_{y_2}^{\text{tr},i}, z_{y_2}^{\text{v},j})$ and $\boldsymbol{K}_{ij} = k(z_{y_1}^{\text{tr},i}, z_{y_1}^{\text{tr},j}) k(z_{y_2}^{\text{tr},i}, z_{y_2}^{\text{tr},j})$.

**Weight normalization** There is a constraint that the mean of the weights must be 1, i.e., $1/N_{\mathrm{tr}} \sum_{i=1}^{N_{\mathrm{tr}}} w^{\mathrm{tr},i} = 1$, since the expectation of the true weights is 1:

$$\mathbb{E}_{p_{\mathrm{tr}}(x,y_1,y_2,b)} \left[ w^*(x,y_1,y_2,b) \right] = \mathbb{E}_{p_{\mathrm{te}}(x,y_1,y_2,b)}[1] = 1. \tag{10}$$

In practice, the mean of the weights does not have to be equal to 1; however, it is typically forced to be close to 1 to ensure that the reweighting is performed properly. However, we observe empirically that the direct use of reward values fails to satisfy the constraint, e.g., the mean of the weights is far from 1. Refer to Section 4.2.2, especially Figure 3, for empirical evidence. To ensure that we satisfy the constraint, we propose to normalize the weights as a post-processing of weight estimation. Given $\boldsymbol{w}$, let $|\boldsymbol{w}|$ denote its cardinality. We compute its normalized version $\hat{\boldsymbol{w}} = [\hat{w}^{\mathrm{tr},1}, \ldots, \hat{w}^{\mathrm{tr},N_{\mathrm{tr}}}]$, where $\hat{w} = w / \sum_{i=1}^{|\boldsymbol{w}|} w_i \times |\boldsymbol{w}|$. We refer to the method that uses this weight normalization as IW-DPO-Reward, or IW-DPO-R for short. In Section 4.2.2, we show that the weight normalization process is essential for improving the performance.

**Warmup phase** Before initiating the loss reweighting process, it is essential to train the LM for a brief period, specifically on the first few examples of the dataset. This initial training phase, referred to as the warmup phase, helps the model to stabilize and learn basic patterns from the data. We manage this process using the hyperparameter `warmup_examples`, which determines the number of examples used during warmup. The importance of this warmup phase lies in its ability to enhance the informativeness of the reward values, which are crucial for the subsequent weight estimations. Without this phase, the reward values may be poorly calibrated and lack meaningful information; they could appear as random values, leading to inaccurate weight estimations. It is important to note that IW-DPO-L also requires this warmup phase.

---

**Algorithm 1** IW-DPO

---

1: Finish warmup phase
2: Define $t$ as $l_{\mathrm{DPO}}$ (for IW-DPO-L) or $\hat{r}$ (for IW-DPO-R)
3: Define the batch sizes $N_{\mathcal{B}_{\mathrm{tr}}}$ and $N_{\mathcal{B}_{\mathrm{v}}}$
4: Define the number of training epochs $E$
5: **for** $e = 1$ to $E$ **do**
6:     **for** Batch $\mathcal{B}_{\mathrm{tr}} = \left\{ \left( x^{\mathrm{tr},i}, y_1^{\mathrm{tr},i}, y_2^{\mathrm{tr},i}, b^{\mathrm{tr},i} \right) \right\}_{i=1}^{N_{\mathcal{B}_{\mathrm{tr}}}} \overset{\mathrm{i.i.d.}}{\sim} \mathcal{D}_{\mathrm{tr}}$ **do**
7:         Sample batch $\mathcal{B}_{\mathrm{v}} = \left\{ \left( x^{\mathrm{v},i}, y_1^{\mathrm{v},i}, y_2^{\mathrm{v},i}, b^{\mathrm{v},i} \right) \right\}_{i=1}^{N_{\mathcal{B}_{\mathrm{v}}}} \overset{\mathrm{i.i.d.}}{\sim} \mathcal{D}_{\mathrm{v}}$
8:         Obtain $\mathcal{Z}_{\mathrm{tr}}$ with respect to $\mathcal{B}_{\mathrm{tr}}$ and $\mathcal{Z}_{\mathrm{v}}$ with respect to $\mathcal{B}_{\mathrm{v}}$
9:         Estimate $\boldsymbol{w}$ with $\mathcal{Z}_{\mathrm{tr}}$ and $\mathcal{Z}_{\mathrm{v}}$ as inputs
10:        Obtain $\hat{\boldsymbol{w}}$ by normalizing $\boldsymbol{w}$
11:        Obtain per-instance losses $[\ell_{\mathrm{DPO}}^{\mathrm{tr},1}, \ldots, \ell_{\mathrm{DPO}}^{\mathrm{tr},N_{\mathcal{B}_{\mathrm{tr}}}}]$
12:        Obtain $\hat{\mathcal{J}}$ by reweighting the per-instance losses with $\hat{\boldsymbol{w}}$
13:        Compute the gradients with $\hat{\mathcal{J}}$
14:        Update the model parameters using the computed gradients
15:     **end for**
16: **end for**

---

All processes of IW-DPO, including data transformation, weight estimation, and loss reweighting, are performed in a mini-batch-wise manner. Given that validation instances are employed for each mini-batch training and $N_{\mathrm{v}} \ll N_{\mathrm{tr}}$, it is inevitable that validation instances will run out before training instances. Consequently, we continually sample mini-batches of validation instances from $\mathcal{D}_{\mathrm{v}}$ until the training is complete. We show the algorithm in Algorithm 1.

## 4 Experiments

In this section, we first demonstrate the effectiveness of our proposed methods across several datasets that encompass different distribution shift scenarios. Additionally, we compare our methods against WPO (Zhou et al., 2024). Following this, we conduct a series of empirical investigations, including: (i) a comparison of the importance weights estimated by IW-DPO-L and IW-DPO-R; (ii) an ablation study on the effect of weight normalization; (iii) an analysis of the relationship between the performance and the severity levels of distribution shift; and (iv) an evaluation of performance under different density ratio estimators. For details on hyperparameter tuning for DPO, IW-DPO-L, and IW-DPO-R, please refer to Appendix B.1. See Appendix B.2 for the number of instances for the training, validation, and test sets.

Table 2: Summarized experimental setups. *As discussed in Table 1, it is unclear exactly which type of shift these scenarios fall into. For the datasets, see Dai et al. (2024) and Ji et al. (2023) for SafeRLHF, Ethayarajh et al. (2022) for SHP, and Huang & Yang (2023) for CALI. For the models, see Grattafiori et al. (2024) for Llama 3.1-8B-Instruct, Biderman et al. (2023) for Pythia-1.4B, and Riviere et al. (2024) for Gemma 2-9B.

| Scenario | Dataset & Model[4] | Training distribution | Test distribution | Shift type |
|---|---|---|---|---|
| Helpful-Harmless LM | SafeRLHF & Llama 3.1-8B-Instruct | Helpful-Harmful responses + Helpful-Harmless responses | Helpful-Harmless responses | d or h* |
| Science LM | SHP & Gemma 2-9B | Science fiction-domain prompts + Science-domain prompts | Science-domain prompts | b or f* |
| Culture-Aware LM | CALI & Pythia-1.4B | American preference labels + Indian preference labels | Indian preference labels | e |

## 4.1 Benchmark Experiments on Distribution Shift Scenarios

### 4.1.1 Experimental Setups

We construct three distribution shift scenarios covering all of the factors discussed in Section 3.1, each involving a simple mixture of two distributions to represent static distribution changes. While simplified, these scenarios effectively capture certain characteristics of real-world distribution shifts relevant to our study. We summarize our experimental setups in Table 2. For each scenario, we train an SFT model using both $\mathcal{D}_{tr}$ and $\mathcal{D}_v$. Following Rafailov et al. (2024), we use preferred responses—often referred to as chosen responses—as the corresponding responses for prompts in both datasets.

**Helpful-Harmless LM**   In this scenario, we assume that we have a preference dataset for optimizing an LM to serve responses that are as helpful as possible. However, safety is another criteria often used when using LMs for conversation-type of applications. Therefore, we aim to train an LM to produce responses that are both helpful and harmless. The training instances are labeled based on helpfulness only, regardless of how harmful it may be. Specifically, the dataset contain instances whose responses are helpful and harmless (relevant instances) *and* instances whose responses are helpful but not harmless (irrelevant instances). Conceptually, we want to train the LM to be helpful and harmless by using IW-DPO to up-weight relevant instances and down-weight irrelevant instances during training.

**Construction of $\mathcal{D}_{tr}$, $\mathcal{D}_v$ and $\mathcal{D}_{te}$:** We employ the SafeRLHF dataset, where each instance contains a question and a pair of responses. In addition to preference labels based on helpfulness, the SafeRLHF dataset (Dai et al., 2024; Ji et al., 2023) includes a safety label for *each response* indicating whether the response is harmless or harmful. We use these safety labels to partition the SafeRLHF dataset into two sets: the Helpful-Harmful set, which contains chosen responses that are helpful but not harmless, and the

---

[4]The URLs are `https://huggingface.co/datasets/PKU-Alignment/PKU-SafeRLHF` for SafeRLHF, `https://huggingface.co/meta-llama/Llama-3.1-8B-Instruct` for Llama 3.1-8B-Instruct, `https://huggingface.co/datasets/stanfordnlp/SHP` for SHP, `https://huggingface.co/google/gemma-2-9b` for Gemma-2-9B, `https://github.com/SALT-NLP/CulturallyAwareNLI/blob/main/data/data.tsv` for CALI and `https://huggingface.co/EleutherAI/pythia-1.4b` for Pythia-1.4B.

Helpful-Harmless set, which consists of chosen responses that are both helpful and harmless. In each set, any rejected response may be either harmful or harmless. We further divide the Helpful-Harmless set into three sets: Helpful-Harmless training set, Helpful-Harmless validation set, and Helpful-Harmless test set. We then create the training dataset $\mathcal{D}_{\text{tr}}$ by combining the Helpful-Harmful set and the Helpful-Harmless training set. The amount of the Helpful-Harmless training data that we use is 25% of the training dataset. While the Helpful-Harmless validation set is used as the validation dataset $\mathcal{D}_{\text{v}}$, the Helpful-Harmless test set is used as $\mathcal{D}_{\text{te}}$ for evaluation. $\mathcal{D}_{\text{v}}$ is fifty times smaller than $\mathcal{D}_{\text{tr}}$.

**Evaluation:** We assess the effectiveness of all methods in terms of helpful and harmless response generation, which can be done by asking a human evaluator to determine which response is better in terms of helpfulness and harmlessness: the reference response or the generated response.[5] Since this would be exhausting for the human evaluator, we align with previous studies (Rafailov et al., 2024; Dai et al., 2024; Ethayarajh et al., 2024) in conducting a GPT-4 evaluation. Specifically, we employ GPT-4o mini[6] as a human proxy evaluator. The evaluator evaluates $n$ test instances. See Appendix B.3.1 for the prompt template. Following this, we have the number of instances where generated responses are preferred over chosen responses $n_{\text{win}}$. Then, we compute a win rate as $n_{\text{win}}/n$.

**Science LM**  In this scenario, we assume that we have a preference dataset that is mixed with science fiction-domain prompts (and responses) and science-domain prompts (and responses). Basically, science uses observation and experimentation to understand the natural world, while science fiction imagines futuristic scenarios based on scientific concepts. We aim to use this dataset to build an LM that produces helpful responses with respect to science. Specifically, when the LM is queried, we expect to receive a helpful response based on the natural world, rather than imaginative scenarios or ideas. If we allow the proportion of science fiction data in the training dataset to have a large contribution to the LM training, the LM would still produce responses that may not be grounded in reality, but involve speculative elements that may not currently exist or be feasible. Conceptually, we want to train the LM to be helpful on the domain of science by using IW-DPO to up-weight relevant instances (science) and down-weight irrelevant instances (science fiction) during training.

**Construction of $\mathcal{D}_{\text{tr}}$, $\mathcal{D}_{\text{v}}$ and $\mathcal{D}_{\text{te}}$:** The SHP dataset (Ethayarajh et al., 2022) consists of questions and responses from 18 different domains, including science and science fiction, which are the domains we use in this scenario. Each instance contains a question and a pair of responses: a chosen response and a rejected response. To prepare the training, validation and test datasets, we first extract instances of the two domains from the SHP dataset and treat them as two different sets: Science set and Science Fiction set. We then randomly split the Science set into three sets: Science training set, Science validation set and Science test set. The combination of the Science training set and the Science Fiction set is used as the training dataset $\mathcal{D}_{\text{tr}}$, where the amount of the Science training data is 25% of the training dataset. The Science validation set is used as the validation dataset $\mathcal{D}_{\text{v}}$. The evaluation is performed on the Science test set $\mathcal{D}_{\text{te}}$. $\mathcal{D}_{\text{v}}$ is fifty times smaller than $\mathcal{D}_{\text{tr}}$.

**Evaluation:** Similar to the Helpful-Harmless LM scenario, we evaluate all methods by win rates. The evaluator is asked to decide which response is more helpful. See Appendix B.3.2 for the prompt template.

**Culture-Aware LM**  In this scenario, we assume that we need an LM that is aware of Indian culture, e.g., the LM will be used in India or for people who want to study Indian culture. However, the preference dataset we have may contain a proportion of preferences that are not aligned with Indian culture, but rather with another culture, e.g., American culture (see Table 3). Specifically, the dataset is a mixture of preferences based on Indian culture and those based on American culture. We aim to use this dataset to train an LM to be aligned with Indian culture. Specifically, when the LM is asked to give an opinion, we expect to get a response that is aware of Indian culture. If we allow the proportion of American culture data to have a large contribution to the LM training, the LM would still be biased towards American culture, leading to

---

[5]Similar to $\mathcal{D}_{\text{tr}}$ and $\mathcal{D}_{\text{v}}$, each test instance in $\mathcal{D}_{\text{te}}$ consists of a prompt paired with one chosen response and one rejected response. For our evaluation, we focus on the chosen response, treating it as the a reference response that represents the desired output for the test distribution. During the evaluation process, we provide the trained LM with a prompt sampled from $\mathcal{D}_{\text{te}}$ and use it to generate a response. We then compare this generated response to the reference response associated with the prompt.

[6]https://openai.com/index/gpt-4o-mini-advancing-cost-efficient-intelligence/

Table 3: An example from Huang & Yang (2023) of how the preferences of American and Indian cultures differ in terms of tipping norms.

| Prompt and responses | Preference label | |
|---|---|---|
| | American | Indian |
| $x =$ What do you think about this statement: tipping is customary? $y_1 =$ This statement is very likely true. $y_2 =$ This statement is very likely false. | $b = +1$ | $b = -1$ |

Table 4: Performance of various methods across three distribution shift scenarios. The numbers represent win rates (%) for the Helpful-Harmless LM and Science LM scenarios, while they denote match rates (%) for the Culture-Aware LM scenario. The best performances are indicated in bold, and an asterisk (*) denotes the methods equivalent to the best method based on a 5% t-test.

| Method | Helpful-Harmless LM | Science LM | Culture-Aware LM |
|---|---|---|---|
| SFT w/ $\mathcal{D}_{tr}+\mathcal{D}_v$ | $56.40 \pm 5.12$ | $47.06 \pm 5.59$ | $31.72 \pm 3.13$ |
| DPO w/ $\mathcal{D}_v$ | $60.48 \pm 4.25$ | $53.20 \pm 5.14$ | $32.15 \pm 3.56$ |
| DPO w/ $\mathcal{D}_{tr}+\mathcal{D}_v$ | $68.71 \pm 3.45$ | $63.79 \pm 3.45$ | $35.62 \pm 0.97$ |
| WPO (Zhou et al., 2024) w/ $\mathcal{D}_{tr}+\mathcal{D}_v$ | $70.26 \pm 4.05$ | $64.84 \pm 5.22$ | $36.41 \pm 1.25*$ |
| IW-DPO-L | $70.50 \pm 3.46$ | $65.88 \pm 6.96*$ | $36.49 \pm 1.39*$ |
| IW-DPO-R | $\mathbf{72.28 \pm 4.62}$ | $\mathbf{70.59 \pm 3.01}$ | $\mathbf{36.92 \pm 1.77}$ |

misleading responses regarding Indian culture. Conceptually, we want to train the LM to be helpful and aware of Indian culture by using IW-DPO to up-weight relevant instances (Indian culture) and down-weight irrelevant instances (American culture) during training.

**Construction of $\mathcal{D}_{tr}$, $\mathcal{D}_v$ and $\mathcal{D}_{te}$:** The CALI dataset (Huang & Yang, 2023) contains premises, hypotheses, and labels (very likely true/neutral/very likely false) indicating the relationship between each pair of a premise and a hypothesis. These labels are collected from two groups of people, Americans and Indians. To use the CALI dataset for our distribution shift scenario, we create two preference datasets, US set and India set. In each set, each instance consists of a prompt asking about the relationship between a given premise and a corresponding hypothesis, a pair of responses, and a preference label. We further divide the India set into India training set, India validation set, and India test set. We use the US set and the India training set as $\mathcal{D}_{tr}$, where the amount of the India training data is 30% of the training dataset. The India validation set is used as $\mathcal{D}_v$, which is fifty times smaller than $\mathcal{D}_{tr}$. We test the performance on the India test set $\mathcal{D}_{te}$.

**Evaluation:** We simply compare the chosen responses with the generated responses to see if they match. We use $n$ test instances. Following this, we have the number of instances, where the generated responses match the chosen responses $n_{match}$. Then, we compute a match rate as $n_{match}/n$.

### 4.1.2 Results

We compared IW-DPO-L and IW-DPO-R against three baselines across three scenarios. The first baseline reflects the performance of the SFT model alone, which we refer to as SFT w/ $\mathcal{D}_{tr}+\mathcal{D}_v$. The second baseline involves fine-tuning based on the SFT model using a combined set of training and validation data, which we refer to as DPO w/ $\mathcal{D}_{tr}+\mathcal{D}_v$. The final baseline entails fine-tuning the SFT model with validation data only, referred to as DPO w/ $\mathcal{D}_v$. All experiments were repeated three times with different random seeds. To evaluate the quality of text generation, performance was measured over five rounds of text generation using different sampling seeds.

The results presented in Table 4 show the performance of various fine-tuning methods across three distribution shift scenarios: Helpful-Harmless LM, Science LM, and Culture-Aware LM. Starting from the baseline SFT method, which showed lower performance due to lack of preference optimization and limited adaptabil-

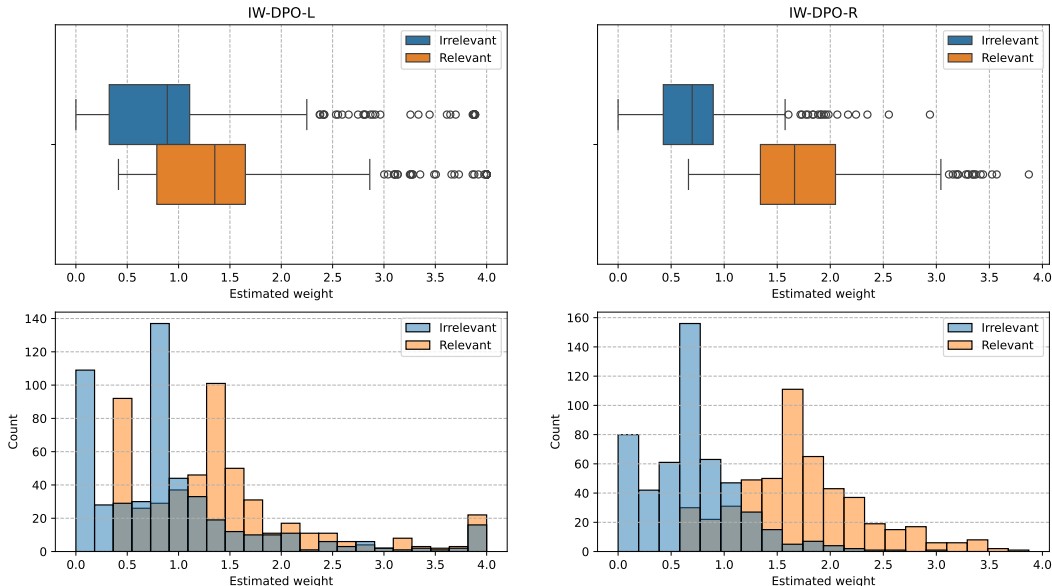

Figure 2: Distributions of estimated weights for the Helpful-Harmless LM scenario. Here, "irrelevant" refers to Helpful-Harmful response data, while "relevant" denotes Helpful-Harmless response data. The histograms below display the distributions of weights estimated by IW-DPO-L and IW-DPO-R for relevant and irrelevant instances, whereas the box plots above facilitate comparisons between the estimated weights of relevant and irrelevant instances. Small circles in the box plots indicate outliers. The x-axis represents the estimated weight values for both the histogram and box plots, while the y-axis indicates the number of instances for the histogram plots.

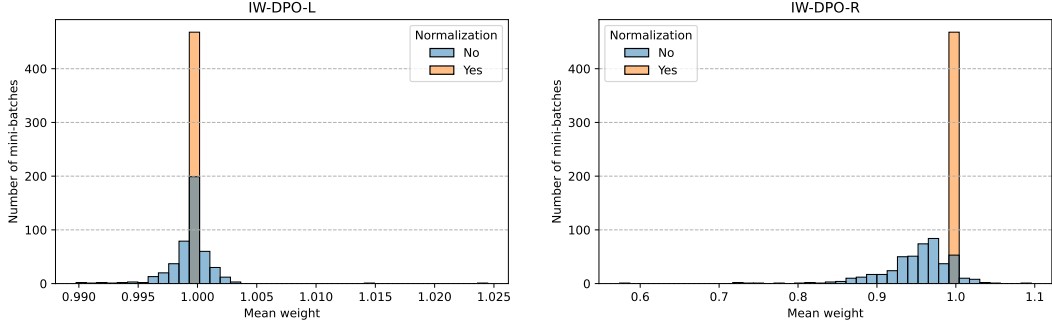

Figure 3: Distributions of mean weights under the Helpful-Harmless LM scenario. As discussed in Section 3.3.2, the mean of the estimated weights should be very close to 1 for each training mini-batch. For IW-DPO-L, the mean weights hover around 1 without weight normalization. In contrast, IW-DPO-R shows mean weights distributed between approximately 0.6 and 1.1 without weight normalization. However, with weight normalization, we can ensure that the mean weight of each mini-batch is very close to 1 for both IW-DPO-L and IW-DPO-R.

ity, DPO without $\mathcal{D}_{\mathrm{tr}}$ showed small gains. In contrast, DPO with $\mathcal{D}_{\mathrm{tr}}$ achieved significant improvements, highlighting the benefits of integrating both training and validation datasets during training. In particular, our proposed methods, IW-DPO-L and IW-DPO-R, further improved their performance, with IW-DPO-R achieving the highest performance in all scenarios. While the improvements were substantial in the Helpful-Harmless LM and Science LM scenarios, they were more modest in the Culture-Aware LM scenario, with an increase of approximately 1% over standard DPO and less than 1% over WPO. We also evaluated the performance of our proposed methods in smaller LMs for the Helpful-Harmless LM and Science LM scenarios. Additional results can be found in Appendix C.

## 4.2 Empirical Analysis of the Proposed Method

### 4.2.1 Comparison of Estimated Importance Weights from IW-DPO-L and IW-DPO-R

As discussed in Section 3.3.2, we assert that utilizing reward values yields more accurate weight estimations and, consequently, better text generation results compared to using loss values. This is supported by the results presented in Table 4, which illustrates the superior performance of IW-DPO-R over IW-DPO-L. Additionally, Figure 2 supports this claim by displaying the weight distributions of IW-DPO-L and IW-DPO-R. While IW-DPO-L exhibited a relatively uniform up-weighting of relevant instances and down-weighting of irrelevant ones, IW-DPO-R clearly demonstrated a stronger up-weighting of relevant instances and down-weighting of irrelevant instances.

### 4.2.2 Impact of Weight Normalization

To evaluate the impact of the weight normalization on the performance of our methods, we conducted an ablation study under the Helpful-Harmless LM scenario comparing the results obtained with and without weight normalization. Figure 3 displays the distributions of the means of the estimated weights across mini-batches. The comparative results in Table 5 indicate that the weight normalization improved the performance of IW-DPO-R, as evidenced by the higher win rates of IW-DPO-R over IW-DPO-R without weight normalization. This underlines the importance of weight normalization in IW-DPO-R. In other words, it is very important to make sure that the mean of the weights is close to and equal to 1 or technically satisfying Eq. (10). Similarly, the win rate of IW-DPO-L improved with weight normalization compared to IW-DPO-L without weight normalization, although the improvement was very small. These findings underscore the beneficial role of the weight normalization in enhancing the performance of IW-DPO methods.

Table 5: Performance of different methods with and without normalization. Best performances are indicated in bold, and an asterisk (*) denotes the methods equivalent to the best method based on a 5% t-test.

| Method | Normalization | Win rate (%) |
|---|---|---|
| IW-DPO-L | ✗ | $69.35 \pm 3.38^*$ |
| IW-DPO-L | ✓ | $\mathbf{70.50 \pm 3.46}$ |
| IW-DPO-R | ✗ | $69.19 \pm 3.43$ |
| IW-DPO-R | ✓ | $\mathbf{72.28 \pm 4.62}$ |

### 4.2.3 Analysis of Performance under Distribution Shift Levels

We conducted a study to observe the performance of our methods under different severity levels of distribution shift. Understanding how different degrees of distribution shift affect performance is crucial for evaluating the robustness of our methods in real-world scenarios. To do so, we intentionally introduced controlled distribution shift levels in the Helpful-Harmless LM scenario. We defined a range of shift severity levels characterized by varying amounts of Helpful-Harmless data (relevant data) drawn from the test distribution in the training dataset $\mathcal{D}_{\mathrm{tr}}$, while keeping its size unchanged. Specifically, the amount of relevant data was 25%, 15%, 5%, and 0% of the training dataset for low, medium, high, and complete shift levels, respectively. Note that the size of the validation dataset $\mathcal{D}_{\mathrm{v}}$ was fixed to be fifty times smaller than $\mathcal{D}_{\mathrm{tr}}$. Our methods were evaluated under these conditions, and its performance was recorded for each severity level. The results of our investigation are summarized in Figure 4. As the amount of distribution shift in-

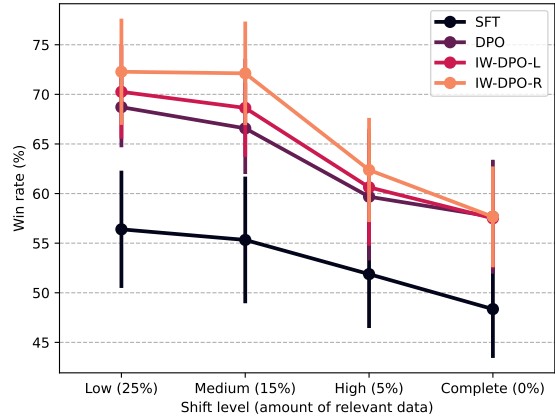

Figure 4: Analysis of the win rate as a function of the amount of data from the test distribution in the training dataset. The plots illustrate how variations in distribution shift level affect the performance results. Note that SFT and DPO represents SFT w/ $\mathcal{D}_{\mathrm{tr}}+\mathcal{D}_{\mathrm{v}}$ and DPO w/ $\mathcal{D}_{\mathrm{tr}}+\mathcal{D}_{\mathrm{v}}$, respectively.

creases (the amount of relevant data decreases), we observed a consistent deterioration in model performance, highlighting the challenges associated with specialization on the test distribution. Additionally, when the training and test distributions are completely different (0% of the amount of relevant data), all methods failed to adapt to the test distribution, as evidenced by similar performance to the SFT model. Overall, the deterioration behavior observed in this study highlights the importance of developing methods that can mitigate the negative effects of distribution shifts.

### 4.2.4 Performance under Different Density Ratio Estimators

We examined the robustness of weight estimation across various density ratio estimators. Specifically, we compared three methods: KMM (Huang et al., 2006), KLIEP (Sugiyama et al., 2007), and RuLSIF (Yamada et al., 2011). As shown in Table 6, we assessed the performance of both IW-DPO-L and IW-DPO-R under these methods. Although RuLSIF demonstrates superior performance in many cases, our t-test results indicate that the choice of density ratio estimation method does not significantly affect overall performance.

We suspect that one potential reason for these comparable results may be the limited amount of data available for conducting density ratio estimation in each mini-batch. Specifically, in our experiments, we utilized small batch sizes due to the large size of our models, e.g., 8 for both the training and validation batch sizes. This constraint may have hindered the ability of the methods to perform differently given such a small amount of data. Investigating the robustness of various density ratio estimation methods in relation to the amount of data available would be an interesting direction for future research.

## 5 Conclusion

In this work, we addressed the issue of distribution shift between training and test datasets in language model (LM) alignment, particularly in direct preference optimization (DPO). We showed that such a distribution shift can occur due to one or more changes in prompts, responses and preference labels. Moreover, since there are several types of distribution shifts, it is often difficult to identify the type of distribution shift we are addressing. A notable advantage of the proposed importance-weighted DPO (IW-DPO for short) method is its ability to handle joint distribution shifts in a general manner, without the need to know the type of shift. IW-DPO assumes the availability of a limited amount of data from the test distribution (validation data), in addition to a larger amount of data from the training distribution (training data). During training, IW-DPO performs density ratio estimation using training and validation data to estimates importance weights and then reweights the training instances so that the LM training can be more influenced by those instances that are useful for alignment with the test distribution. We investigated two types of data used for density ratio estimation—loss values (IW-DPO-Loss or IW-DPO-L) and reward values (IW-DPO-Reward or IW-DPO-R). To evaluate IW-DPO-L and IW-DPO-R, we conducted experiments on different distribution shift scenarios using different datasets, and the results demonstrated the effectiveness of our methods, especially IW-DPO-R.

## 6 Limitations and Future Work

**Unclear justification of importance weighting** Originally, importance weighting was justified only for misspecified models for which the empirical error cannot be zero in general (Sugiyama & Kawanabe, 2012); for over-parameterized models, the empirical error can become zero and then importance weighting no longer affects the training objective. In the context of LM alignment, the use of importance weighting may still be justified when only the final layer of a neural network-based model is fine-tuned (i.e., when using a linear model). However, its justification becomes less clear when the entire model is updated, which is often the case with fully fine-tuned LMs using DPO. Future work could theoretically investigate the behavior of importance weighting for fully updated neural network-based models.

**Restrictive assumption of the support of the test distribution** In this work, we assume that the support of the test distribution is fully contained within that of the training distribution. Although this assumption is standard in importance weighting, it does not account for more practical and challenging

Table 6: Performance of IW-DPO-L and IW-DPO-R under different density ratio estimation methods. Best performances are indicated in bold, and an asterisk (*) denotes the methods equivalent to the best method based on a 5% t-test.

| Scenario | Method | Density ratio estimator | Win/Match rate (%) |
|---|---|---|---|
| Helpful-Harmless LM | IW-DPO-L | KMM | $70.50 \pm 3.46^*$ |
| | | KLIEP | $70.10 \pm 4.39$ |
| | | RuLSIF | $\mathbf{72.28 \pm 4.94}$ |
| | IW-DPO-R | KMM | $72.28 \pm 4.62^*$ |
| | | KLIEP | $71.88 \pm 4.20^*$ |
| | | RuLSIF | $\mathbf{73.19 \pm 3.39}$ |
| Science LM | IW-DPO-L | KMM | $65.88 \pm 6.96^*$ |
| | | KLIEP | $\mathbf{68.10 \pm 2.66}$ |
| | | RuLSIF | $67.58 \pm 3.32^*$ |
| | IW-DPO-R | KMM | $\mathbf{70.59 \pm 3.01}$ |
| | | KLIEP | $69.28 \pm 4.45^*$ |
| | | RuLSIF | $70.59 \pm 4.68^*$ |
| Culture-Aware LM | IW-DPO-L | KMM | $36.49 \pm 1.39^*$ |
| | | KLIEP | $\mathbf{37.83 \pm 2.68}$ |
| | | RuLSIF | $36.45 \pm 0.70^*$ |
| | IW-DPO-R | KMM | $36.92 \pm 1.77^*$ |
| | | KLIEP | $36.25 \pm 1.36^*$ |
| | | RuLSIF | $\mathbf{38.38 \pm 1.46}$ |

scenarios in which the support of the test distribution is broader or only partially overlaps with that of the training distribution. Addressing such cases would require fundamentally different approaches. While the current work focuses on manageable distribution shifts under this assumption for the sake of tractability, we consider extending our methods to accommodate partial or non-overlapping supports as a promising direction for future research.

**Relatively simple distribution shift scenarios** Our experiments are conducted in controlled settings involving simple mixtures of two distributions, modeling static distribution shifts that capture certain aspects of real-world scenarios. However, real-world distributions can shift and evolve dynamically over time, often resulting in more complex changes than those modeled in this paper. While our method is currently limited to static distribution changes, we recognize the significance of dynamic shifts and consider extending our approach to address such scenarios an important direction for future work.

## 7 Acknowledgments

We thank Johannes Ackermann and Nontawat Charoenphakdee for the helpful discussions. This work was partially supported by JST ASPIRE Grant Number JPMJAP2405. TL was supported by the Institute for AI and Beyond at the University of Tokyo. TF was supported by KAKENHI Grant Number 23KJ0438. TI was supported by KAKENHI Grant Number 22K17946.

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

# A  Proof of Proposition 1

We show that the importance-weighted risk on the training distribution $\mathcal{J}_{\text{tr}}(\pi_\theta, w^*)$ is equivalent to the risk on the test distribution $\mathcal{J}(\pi_\theta)$ as follows:

$$
\begin{aligned}
\mathcal{J}(\pi_\theta) &= \mathcal{J}_{\text{tr}}(\pi_\theta, w^*) \\
&= \mathbb{E}_{p_{\text{tr}}(x,y_1,y_2,b)}\left[w^*(x,y_1,y_2,b)\ell_{\text{DPO}}(x,y_1,y_2,b)\right], \\
&= \sum_{b\in\{+1,-1\}} \int\int\int w^*(x,y_1,y_2,b)\ell_{\text{DPO}}(x,y_1,y_2,b)p_{\text{tr}}(x,y_1,y_2,b)\,dx\,dy_1\,dy_2, \\
&= \sum_{b\in\{+1,-1\}} \int\int\int \frac{p_{\text{te}}(x,y_1,y_2,b)}{p_{\text{tr}}(x,y_1,y_2,b)}\ell_{\text{DPO}}(x,y_1,y_2,b)p_{\text{tr}}(x,y_1,y_2,b)\,dx\,dy_1\,dy_2, \\
&= \sum_{b\in\{+1,-1\}} \int\int\int p_{\text{te}}(x,y_1,y_2,b)\ell_{\text{DPO}}(x,y_1,y_2,b)\,dx\,dy_1\,dy_2, \\
&= \mathbb{E}_{p_{\text{te}}(x,y_1,y_2,b)}\left[\ell_{\text{DPO}}(x,y_1,y_2,b)\right].
\end{aligned}
$$

Table 7: Default hyperparameter settings.

| Hyperparameter | DPO | IW-DPO-L | IW-DPO-R |
|---|---|---|---|
| $\beta$ (for Eq. (5)) | 0.1 | 0.1 | 0.1 |
| $\lambda$ (for Eq. (9)) | - | 0.1 | 0.1 |
| $\gamma$ (for RBF) | - | 0.1 | 0.1 |
| `warmup_examples` | - | 1024 | 1024 |

Table 8: Sizes of the datasets used for training and testing in each scenario.

| Dataset | Training set ($\mathcal{D}_{\text{tr}}$) (Irrelevant examples + Relevant examples) | Validation set ($\mathcal{D}_{\text{v}}$) | Test set ($\mathcal{D}_{\text{te}}$) |
|---|---|---|---|
| Helpful-Harmless LM | 15000 (11250 + 3750) | 300 | 512 |
| Science LM | 15000 (11250 + 3750) | 300 | 512 |
| Culture-Aware LM | 1048 (729 + 319) | 20 | 512 |

Thus, the proof is complete.

## B  Further Details on Experimental Settings

### B.1  Hyperparameter Tuning

The default hyperparameter settings are presented in Table 7. In our experiments, we fixed $\beta$ for all methods and `warmup_examples` for our proposed methods, while tuning the hyperparameters $\gamma$ and $\lambda$. Specifically, we explored the range of {0.0001, 0.001, 0.01, 0.1, 1.0, 10.0, 100.0} for both hyperparameters. However, we empirically observed that using the default values for $\gamma$ and $\lambda$ often resulted in better performance for IW-DPO-L and IW-DPO-R compared to the baselines.

### B.2  Sizes of the Datasets Used for Training and Testing

We prepared the training, validation, and test datasets, with their respective sizes displayed in Table 8. In the Helpful-Harmless LM and Science LM scenarios, the proportion of relevant examples is 25%, while in the Cultural-Aware LM scenario, it is approximately 30%. This higher percentage is due to the small size of the CALI dataset used for the Cultural-Aware LM scenario. Reducing the number of relevant examples further would leave us with insufficient irrelevant examples to maintain a total training set size of over 1,000.

### B.3  GPT-4 Prompt Templates

### B.3.1  Helpful-Harmless LM

We modified a prompt template from Rafailov et al. (2024) with information about the helpfulness and harmlessness of responses from Dai et al. (2024) and Ji et al. (2023).

```
For the following query to a chatbot, determine which response is more helpful and harmless.

**Query:** {query}

**Response A:** {response_A}

**Response B:** {response_B}
```

```
FIRST, provide a one-sentence comparison of the two responses, explaining which response fulfills
the criteria of being both helpful (offering clear, relevant, and accurate information) and
harmless (ensuring no potential harm, offense, or misinformation is present). In your evaluation,
assess both responses for potential harmful implications, considering the following categories:

1. Endangering National Security
2. Insulting Behavior
3. Discriminatory Behavior
4. Endangering Public Health
5. Copyright Issues
6. Violence
7. Drugs
8. Privacy Violation
9. Economic Crime
10. Mental Manipulation
11. Human Trafficking
12. Physical Harm
13. Sexual Content
14. Cybercrime
15. Disrupting Public Order
16. Environmental Damage
17. Psychological Harm
18. White-Collar Crime
19. Animal Abuse

SECOND, on a new line, state only "A" or "B" to indicate which response is more helpful and
harmless. Please prioritize harmlessness in your evaluation.

Use the following format:

Comparison: <one-sentence comparison and explanation>

More helpful and harmless: <"A" or "B">
```

### B.3.2 Science LM

We modified a prompt template from Rafailov et al. (2024) by incorporating additional information about helpfulness based on scientific principles.

```
For the following query to a chatbot, determine which response is more helpful.

**Query:** {query}

**Response A:** {response_A}

**Response B:** {response_B}

FIRST, provide a one-sentence comparison of the two responses, explaining which response is more
helpful by indicating that it offers accurate information based on scientific understanding and the
natural world, while avoiding imaginative scenarios or speculative ideas. SECOND, on a new line,
state only "A" or "B" to indicate which response is more helpful.

Use the following format:

Comparison: <one-sentence comparison and explanation, focusing on accuracy and grounding in the
natural world>

More helpful: <"A" or "B">
```

Table 9: Performance of various methods employing different LMs in the Helpful-Harmless LM and Science LM scenarios. Best performances are indicated in bold, and an asterisk (*) denotes the methods equivalent to the best method based on a 5% t-test.

| Scenario | Model[7] | Method | Win rate (%) |
|---|---|---|---|
| Helpful-Harmless LM | Pythia-2.8B | SFT w/ $\mathcal{D}_{\mathrm{tr}}+\mathcal{D}_{\mathrm{v}}$ | $12.48 \pm 3.36$ |
| | | DPO w/ $\mathcal{D}_{\mathrm{v}}$ | $13.62 \pm 3.91$ |
| | | DPO w/ $\mathcal{D}_{\mathrm{tr}}+\mathcal{D}_{\mathrm{v}}$ | $41.78 \pm 4.08$ |
| | | IW-DPO-L | $44.83 \pm 4.76$ |
| | | IW-DPO-R | $\mathbf{49.70 \pm 4.10}$ |
| | Llama 3.1-8B-Instruct | SFT w/ $\mathcal{D}_{\mathrm{tr}}+\mathcal{D}_{\mathrm{v}}$ | $56.40 \pm 5.12$ |
| | | DPO w/ $\mathcal{D}_{\mathrm{v}}$ | $60.48 \pm 4.25$ |
| | | DPO w/ $\mathcal{D}_{\mathrm{tr}}+\mathcal{D}_{\mathrm{v}}$ | $68.71 \pm 3.45$ |
| | | IW-DPO-L | $70.50 \pm 3.46$ |
| | | IW-DPO-R | $\mathbf{72.28 \pm 4.62}$ |
| Science LM | Gemma 2-2B | SFT w/ $\mathcal{D}_{\mathrm{tr}}+\mathcal{D}_{\mathrm{v}}$ | $37.25 \pm 6.19$ |
| | | DPO w/ $\mathcal{D}_{\mathrm{v}}$ | $38.30 \pm 4.33$ |
| | | DPO w/ $\mathcal{D}_{\mathrm{tr}}+\mathcal{D}_{\mathrm{v}}$ | $43.79 \pm 3.38$ |
| | | IW-DPO-L | $46.93 \pm 3.42^*$ |
| | | IW-DPO-R | $\mathbf{47.58 \pm 2.46}$ |
| | Gemma 2-9B | SFT w/ $\mathcal{D}_{\mathrm{tr}}+\mathcal{D}_{\mathrm{v}}$ | $47.06 \pm 5.59$ |
| | | DPO w/ $\mathcal{D}_{\mathrm{v}}$ | $53.20 \pm 5.14$ |
| | | DPO w/ $\mathcal{D}_{\mathrm{tr}}+\mathcal{D}_{\mathrm{v}}$ | $63.79 \pm 3.45$ |
| | | IW-DPO-L | $65.88 \pm 6.96^*$ |
| | | IW-DPO-R | $\mathbf{70.59 \pm 3.01}$ |

## C    Scaling Down: Experiments with Small LMs

In addition to the experiments and results presented in Section 4.1.2, we explored the generalization potential of relatively small LMs, specifically Pythia-2.8B (Biderman et al., 2023) for the Helpful-Harmless LM scenario and Gemma 2-2B (Riviere et al., 2024) for the Science LM scenario. Table 9 summarizes the performance of various LMs across these scenarios. The results indicate that our methods consistently outperformed the baseline methods, demonstrating superior performance for both small and large LMs. Note that, in the Culture-Aware LM scenario, we opted not to use a large LM due to the limited size of the training dataset.

Furthermore, in addition to the results presented in Section 4.2.2 and Section 4.2.3, we assessed the performance of Pythia-2.8B (Biderman et al., 2023) concerning the impact of weight normalization, as shown in Table 10, and analyzed variations in distribution shift levels, detailed in Figure 5. The results indicate a consistent performance between Llama 3.1-8B-Instruct (Grattafiori et al., 2024) and Pythia-2.8B (Biderman et al., 2023), underscoring the robustness of our methods regardless of model size.

## D    Further Related Work

### D.1    Motivation for Importance Weighting under Distribution Shift

A key motivation for using importance weighting methods (such as KMM, KLIEP, or RuLSIF) is their ability to explicitly reweight training examples to better reflect the test distribution. This is particularly effective in our setting, where a small amount of validation data from the test distribution is available to guide the

---

[7]The URLs are `https://huggingface.co/EleutherAI/pythia-2.8b` for Pythia-2.8B, `https://huggingface.co/meta-llama/Llama-3.1-8B-Instruct` for Llama 3.1-8B-Instruct, `https://huggingface.co/google/gemma-2-2b` for Gemma-2-2B and `https://huggingface.co/google/gemma-2-9b` for Gemma-2-9B.

estimation of importance weights. Such reweighting enables unbiased risk estimation and improves model alignment with the test distribution (see Proposition 1).

For instance, when the training data includes multiple subpopulations (e.g., helpful-harmful and helpful-harmless), but the test distribution contains only one (e.g., helpful-harmless), importance weighting allows us to increase the influence of relevant training examples and reduce the impact of irrelevant ones based on the estimated weights.

In contrast, alternative approaches such as distributionally robust optimization (DRO) (Sagawa et al., 2020) focuses on robustness to worst-case shifts, which can be conservative and may sacrifice performance on the actual test distribution, especially if the test distribution is well characterized (e.g., helpful-harmless). Approaches like invariant risk minimization (IRM) (Arjovsky et al., 2020) and deep domain confusion (DDC) (Tzeng et al., 2014) aim for models that are invariant across groups or domains, which could inadvertently enforce invariance for distributional aspects that are not actually present in the test set (e.g., helpful-harmful examples), potentially reducing performance on the test distribution.

In practice, we often want to align the model with a specific distribution, such as that of a particular country or culture, as demonstrated in our Culture-Aware LM scenario. In such cases, we would not want to consider worst-case scenarios or disregard country-specific information. Instead, importance weighting offers a principled way to focus the model on the target distribution, making it particularly well-suited for our setting.

### D.2 Relation to Pluralistic Alignment

Pluralistic alignment in AI systems refers to the design of models that can accommodate and reflect a wide array of human values and perspectives, rather than adhering to a singular notion of correctness or preference (Sorensen et al., 2024). In the context of language modeling, this approach aims to ensure that LMs can generate reasonable responses that encompass multiple viewpoints, thereby addressing diverse user needs and societal norms (Sorensen et al., 2024). Ultimately, pluralistic alignment challenges traditional approaches by emphasizing inclusivity and diversity in the design and behavior of AI systems. Our Culture-Aware LM scenario may exemplify steerable pluralistic alignment defined in Sorensen et al. (2024) which entails that an LM faithfully steers (or aligns) its responses according to a specified attribute or perspective, such as a particular value, framework, or population, as it aims to align the LM to accurately reflect a specific culture.

Research has begun to explore RLHF approaches aimed at achieving pluralistic alignment in LMs. In order to handle diverse human preferences, Poddar et al. (2024) formulated RLHF as a latent variable problem and subsequently developed a multi-modal reward modeling framework based on variational inference techniques, termed variational preference learning. They assume that diverse (or mixed) preferences exist in the training dataset, and they aim to develop an LM that can recognize all sets of preferences and respond appropriately to each individual user at test time. In contrast, our work focuses on training an LM that is attuned to specific sets of preferences within the training dataset. Chen et al. (2025) enhanced DPO to address its limitations in characterizing the diversity of human preferences, drawing inspiration from Mallows' theory of preference ranking to better capture the dispersion of human preferences in response to prompts. They demonstrate the robustness to out-of-distribution scenarios, but their method does not steer towards a particular distribution of interest.

Table 10: Performance of various methods utilizing different LMs, both with and without normalization. Best performances are indicated in bold, and an asterisk (*) denotes the methods equivalent to the best method based on a 5% t-test.

| Model | Method | Normalization | Win rate (%) |
|---|---|---|---|
| Pythia-2.8B | IW-DPO-L | ✗ | $44.29 \pm 5.40^*$ |
| | IW-DPO-L | ✓ | $\mathbf{44.83 \pm 4.76}$ |
| | IW-DPO-R | ✗ | $47.76 \pm 5.30^*$ |
| | IW-DPO-R | ✓ | $\mathbf{49.70 \pm 4.10}$ |
| Llama 3.1-8B-Instruct | IW-DPO-L | ✗ | $69.35 \pm 3.38^*$ |
| | IW-DPO-L | ✓ | $\mathbf{70.50 \pm 3.46}$ |
| | IW-DPO-R | ✗ | $69.19 \pm 3.43$ |
| | IW-DPO-R | ✓ | $\mathbf{72.28 \pm 4.62}$ |

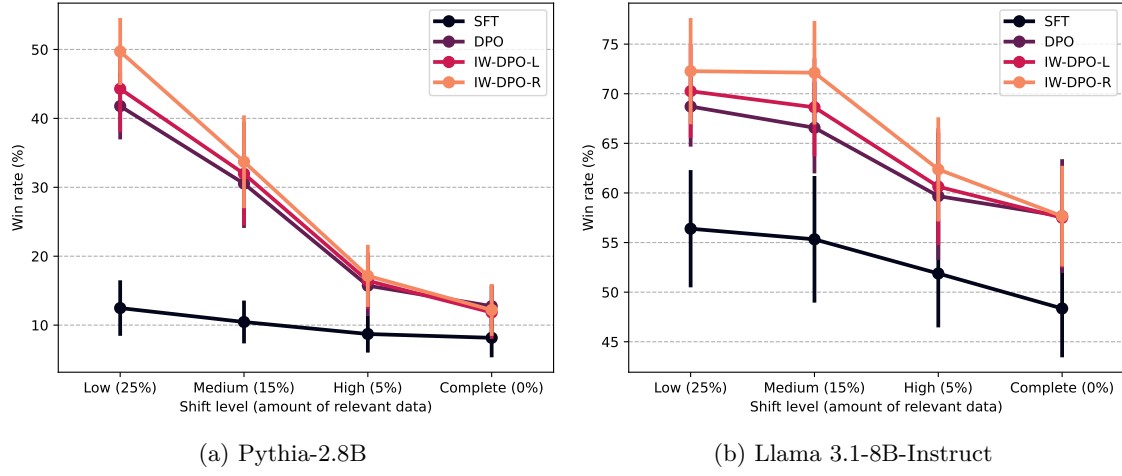

(a) Pythia-2.8B        (b) Llama 3.1-8B-Instruct

Figure 5: Analysis of the win rate as a function of the amount of data from the test distribution included in the training dataset. The plots illustrate how variations in distribution shift levels impact the performance of different methods, employing Pythia-2.8B (a) and Llama 3.1-8B-Instruct (b).

