# OpenReview forum: "Importance Weighting for Aligning Language Models under Deployment Distribution Shift"
_TMLR — Accepted by TMLR_

### Review · Reviewer_NepN · 2025-02-23

**Summary Of Contributions:**

This paper introduces Importance-Weighted Direct Preference Optimization (IW-DPO) to tackle deployment distribution shift in aligning large language models (LLMs) to human preferences. Unlike previous work focusing on model distribution shift or reward modeling, this paper proposes an importance weighting approach to reweight training data based on a small set of validation examples sampled from the test distribution. The authors conduct extensive experiments on different distribution shift scenarios and demonstrate IW-DPO’s superiority over standard DPO baselines.

**Audience:**

Yes

**Broader Impact Concerns:**

- **Bias amplification risk:** If the validation dataset used for importance weighting is biased or unrepresentative, IW-DPO may reinforce and propagate these biases rather than mitigate them.
- **Security vulnerability:** IW-DPO’s reliance on test distribution data makes it susceptible to adversarial manipulation, where malicious actors could inject biased validation examples to steer model behavior.
- **Computational overhead:** The additional reweighting process increases training complexity and environmental costs, raising concerns about scalability and energy efficiency for large-scale language models.

**Claims And Evidence:**

Yes

**Requested Changes:**

1. Provide deeper theoretical justification for why importance weighting benefits DPO.
2. Investigate robustness of weight estimation under different density ratio estimators.
3. Compare IW-DPO against alternative robust optimization baselines, not just standard DPO.
4. Evaluate scaling to larger LMs to understand generalization potential.

**Strengths And Weaknesses:**

Strength:
1. The paper tackles a crucial problem in LLM alignment, deployment distribution shift, which is often overlooked in current RLHF and DPO literature.
2. Introducing importance weighting to reweight preference data in DPO is a valuable extension.
3. Unlike prior methods (e.g., Weighted Preference Optimization, WPO), IW-DPO is more general and does not assume a specific type of distribution shift.
4. Use of multiple datasets (SafeRLHF, SHP, CALI) and multiple LMs (Pythia-2.8B, Gemma 2-2B, Pythia-1.4B) ensures broad applicability.
5. Well-designed ablation studies examining weight normalization, choice of transformation function, and robustness to shift severity.
6. Consistently outperforms standard DPO baselines across all scenarios.

Weakness:
1. The paper should extend the theoretical discussion on how importance weighting affects preference learning dynamics.
2. Kernel Mean Matching (KMM) might not be optimal for estimating density ratios in high-dimensional preference data.
3. Weight normalization improves performance, but how sensitive is IW-DPO to poor weight estimation? Could erroneous weight estimation lead to overfitting or degradation in preference alignment?
4. The paper compares IW-DPO only against standard DPO baselines, but not against other domain adaptation or robust optimization approaches.
5. The paper claims that IW-DPO and WPO are different, but a direct empirical comparison is missing.
6. The study uses relatively small-scale models (Pythia-2.8B, Gemma 2-2B).

---

> ### Author Response · Authors · 2025-03-30
> **Reply**
>
> Thank you very much for your review. We appreciate your suggestions and comments. We have responded to your requested changes as follows:
>
> ---
>
> **Q1)** Provide deeper theoretical justification for why importance weighting benefits DPO.
>
> ***A1)*** We have revised Section 3.2.1 to include Proposition 1 that establishes how importance weighting positively impacts DPO and a discussion of why importance weighting is important. Specifically, by utilizing the importance weight $w$, we can ensure that minimizing the risk on the training distribution aligns with minimizing the risk on the test distribution, thereby effectively optimizing performance for the test distribution. In contrast, without employing $w$, there is no assurance that risk minimization on the training distribution will correspond to risk minimization on the test distribution.
>
> ---
>
> **Q2)** Investigate robustness of weight estimation under different density ratio estimators.
>
> ***A2)*** We have incorporated the results of our investigation into weight estimation using different density ratio estimators in Section 4.2.4. Specifically, we compared three estimators: Kernel Mean Matching (KMM) [Huang et al. (2006)], Kullback–Leibler Importance Estimation Procedure (KLIEP) [Sugiyama et al. (2007)], and Relative Unconstrained Least-Squares Importance Fitting (RuLSIF) [Yamada et al. (2011)]. Our findings demonstrate that the choice of density ratio estimation method is not significant.
>
> ---
>
> **Q3)** Compare IW-DPO against alternative robust optimization baselines, not just standard DPO.
>
> ***A3)*** We performed experiments with WPO [Zhou et al. (2024)]. We added the results for WPO in Table 4. Overall, while WPO outperformed the standard DPO, it still fell short of surpassing our IW-DPO.
>
> ---
>
> **Q4)** Evaluate scaling to larger LMs to understand generalization potential.
>
> ***A4)*** We agree with the importance of evaluating larger LMs for assessing generalization potential. In our updated results, we have included evaluations using Llama 3.1-8B-Instruct for the Helpful-Harmless LM scenario and Gemma 2-9B for the Science LM scenario, documented in Section 4, with our earlier results with smaller models relocated to Appendix C. The updated results demonstrate that our method consistently outperformed the baseline methods when applied to these larger LMs. For the Culture-Aware LM scenario, we believe 1.4B remains a suitable choice, given the small size of the training dataset.
>
> ---
>
> References
> - Correcting Sample Selection Bias by Unlabeled Data, Huang et al. (2006)
> - Direct Importance Estimation With Model Selection and Its Application to Covariate Shift Adaptation, Sugiyama et al. (2007)
> - Relative Density-Ratio Estimation for Robust Distribution Comparison, Yamada et al. (2011)
> - WPO: Enhancing RLHF with Weighted Preference Optimization, Zhou et al. (2024)

---

### Review · Reviewer_KNsG · 2025-03-09

**Summary Of Contributions:**

This paper aims to alleviate the effects of distribution shifts in DPO (direct preference optimisation), i,e., the training distribution is significantly different from deployment environment.  Specially, it proposes an important weight term considering a small set of test samples within the DPO training objective. To facilitate the weight estimation, it uses the radial kernel and utilises the reward value for the kernel transformation function. The framework is evaluated under three scenario, i,e., harmful, science and culture, based on two model types, i..e, pythia and gemma, across eight distribution shifts.

**Audience:**

Yes

**Claims And Evidence:**

Yes

**Requested Changes:**

1. Add experimental comparison results with [1]
2. Discuss the relatedness and novelty of your work by comparing with papers in the broader topic "how to adjust training sample weight".

**Strengths And Weaknesses:**

Strengths:
1. The paper is well-written and easy to follow.
2. The analysis of distributions is comprehensive and inspiring.
3. Beyond performance enhancements in Table4, the results shown in Figure1 showing the importance weight does identify the relevant samples further provides evidence about the method strength and insights.

Weakness:
1. The literature review about exactly "distribution shift" in LLM alignment training is missing[1]. And there is no comparison baseline.
2. This targeted area is broadly also related to "how to adjust training sample weight", in which there are lots of existing works[2,3,4].


References:

[1] 	Importance weighting can help large language models self-improve

[2]    DYNAMIC LOSS-BASED SAMPLE REWEIGHTING FOR IMPROVED LARGE LANGUAGE MODEL PRETRAINING

[3]    Adaptive training distributions with scalable online bilevel optimization

---

> ### Author Response · Authors · 2025-03-30
> **Reply**
>
> Thank you very much for your review. We appreciate your suggestions and comments. We have responded to your requested changes as follows:
>
> ---
>
> **Q1)** Add experimental comparison results with [1]
>
> ***A1)*** We are unable to apply the method from [1], as self-consistency and majority voting are typically effective only for datasets or tasks with clear, definitive answers. In RLHF, we encounter more open-ended tasks, making it difficult to perform self-consistency and majority voting. Moreover, the datasets used in our experiments are designed to provide helpful responses, which further limits the applicability of the method from [1]. However, we are thankful for the suggestion and believe it is an important related work. We added discussions about this paper in Section 2.1.
>
> ---
>
> **Q2)** Discuss the relatedness and novelty of your work by comparing with papers in the broader topic "how to adjust training sample weight".
>
> ***A2)*** We have added discussions regarding the differences between our work and the papers [1,2,3] in Section 2.1. Regarding your mention of [4], we noticed that there is no reference labeled as [4] in your references. Could you please provide the title of the paper you are referring to as [4]?

---

> > ### Comment · Reviewer_KNsG · 2025-04-15
> >
> > Thanks for the updated comparison with the literatures I mentioned in Section 2.1.
> >
> > One remaining concern is that, although methods like WPO aim to mitigate distribution shifts between the on-policy and off-policy distributions, the general problem of distribution mismatch is well-studied, and there are many existing and standard approaches that could, in principle, be applicable here. For instance, density ratio estimation, importance weighting, and domain adaptation techniques have been extensively developed in prior work [1,2,3].
> >
> > Therefore, my main concern is the lack of awareness or comparison with these traditional methods for addressing distribution shifts. Kernel estimation (or kernel mean matching) is just one possible solution within the broader space of distribution correction techniques. This critique is aligned with Reviewer NepN’s Weakness 4.
> > I wondered Why was kernel estimation selected in particular? Why not other alternatives such as invariant learning, DRO, or adversarial domain adaptation?
> >
> > A brief discussion or ablation justifying the selection of kernel estimation over other methods would significantly strengthen the empirical and theoretical contributions of the work.
> >
> >
> > WPO: Enhancing RLHF with Weighted Preference Optimization
> >
> > References:
> >
> > [1] Distributionally Robust Neural Networks for Group Shifts: On the Importance of Regularization for Worst-Case Generalization.
> >
> > [2] Invariant Risk Minimization
> >
> > [3] Deep Domain Confusion: Maximizing for Domain Invariance

---

> > > ### Author Response · Authors · 2025-04-22
> > > **Reply**
> > >
> > > Thank you for reading our rebuttal. In our work, we study distribution shifts between the training and test sets, and this type of shift is precisely where our approach excels because IW-DPO (importance weighting with methods such as KMM, KLIEP, or RuLSIF) seeks to directly correct for distribution mismatch by explicitly reweighting training examples to better reflect the test distribution. This enables unbiased risk estimation under the test distribution. Since the training set combines different subpopulations (e.g., helpful-harmful and helpful-harmless), but test-time only sees one subpopulation (e.g., helpful-harmless), we can use available test distribution data (validation data) to estimate weights that increase the contribution of (helpful-harmless) examples likely under the test distribution, while decreasing that of (helpful-harmful) examples unlikely under the test distribution. This direct adjustment is effective for our deployment distribution shift problem.
> > >
> > > On the other hand, DRO [1] focuses on robustness to worst-case shifts, which can be conservative and may sacrifice performance on the actual test distribution, especially if the test distribution is well characterized (e.g., helpful-harmless). IRM [2] and DDC [3] aim for models that are invariant across groups or domains, which could inadvertently enforce invariance for distributional aspects that are not actually present in the test set (such as helpful-harmful examples), potentially reducing performance on the test distribution. Additionally, many unsupervised domain adaptation methods (e.g., the unsupervised setup in DDC) implicitly assume that $p(y \mid x)$ is unchanged between the training and test distributions. This assumption is often violated in our response shift scenarios, which may lead to suboptimal performance.
> > >
> > > In practice, we often want to align the language model to a certain distribution, e.g., to a specific country or culture as shown in one of our experiments. We would not want to consider worst-case shifts or potentially discard country specific information in such cases. Thank you for highlighting these related papers. We will incorporate a discussion of them in our paper.

---

### Review · Reviewer_CaZY · 2025-03-16

**Summary Of Contributions:**

This paper studies the problem of aligning large language models (LLMs) under deployment distribution shift, where training and test preference data used for alignment are from two different distributions. The authors introduce an importance-weighted direct policy optimization (DPO) method, which uses a small amount of validation data from the test distribution to perform distribution matching. This is done by first transforming training and validation data and then using kernel mean matching for weight estimation. The paper evaluates two variants of the proposed method on three semi-synthetic scenarios, which shows better performance in comparison with DPO without importance weighting.

**Audience:**

Yes

**Broader Impact Concerns:**

No particular concerns

**Claims And Evidence:**

Yes

**Requested Changes:**

See above. Additionally,
- Could you provide more details on the experiments? For example, what are the sizes of the datasets used for training and testing in each setting?
- Why does the third setting use 45% test distribution data for training, whereas the first two only incorporate 25%?
- The paper is generally easy to follow. For clarity, in Section 3.2, it would be nice if you could briefly explain the motivation for applying a transformation function $t$. In Section 3.3.2, it would be helpful to refer to equation (5) when discussing reward values. In Section 4.1.1, for evaluation metrics, it is not entirely clear what chosen responses and generated responses mean.
- While related work is discussed in Section 1, a more thorough review would be beneficial, given the extensive research on importance weighting, domain adaptation, and distribution shift in both traditional settings and LLM domains. In addition, how is this problem related to pluralistic alignment [https://pluralistic-alignment.github.io/] -- are there techniques there that can be applied to handle distribution shift?

**Strengths And Weaknesses:**

**Strengths**:
- The paper addresses distribution shift in LLM alignment, which is practically relevant and important for real-world deployment.
- The overall approach is conceptually intuitive and clean.
- The proposed approach can handle various types of distribution shift, including those in prompts, responses and preference labels, without knowledge on the contributing factors.
- The authors evaluated the proposed method on three scenarios, each constructed with a different dataset.

**Weaknesses**:
- If my understanding is correct, the paper assumes that the support of the test distribution is contained within that of the training distribution. This seems quite restrictive -- what if there are some unseen prompts or responses? In addition, what if the learner cannot obtain a representative validation dataset from the test distribution -- can anything still be done here?

- The experiments are in very controlled settings that involve simple mixtures of two distributions. From the results in Table 4, the improvement in win/match rates from using importance weighting seems relatively modest (3 - 8%) -- it is unclear how much benefits they bring in more complex real-world settings. In addition, It is unclear what win/match rates should be considered as meaningful targets, as there is no baseline evaluation using, e.g., DPO on the test distribution. It would also be beneficial to include comparisons with approaches such as [Zhou et al. (2024] or other domain adaptation techniques that do not use importance weighting.

- The current setting assumes a single training distribution and a single test distribution, but in real-world applications, distributions may shift and evolve over time. It would be nice if the proposed approach can be extended to handle dynamic changes.

- This is not a major concern, as novelty is not a necessary criterion. The proposed method is largely built on the techniques of nonlinear transformation of data and kernel mean matching for weight estimation, as introduced in [Fang et al., 2020] and [Huang et al., 2006]. Having said that, the paper does extend these ideas to LLM alignment and introduce the use of implicit reward functions for data transformation.

---

> ### Author Response · Authors · 2025-03-30
> **Reply (part 1)**
>
> Thank you very much for your review. We appreciate your suggestions and comments. We have responded to your pointed weaknesses and requested changes as follows:
>
> ---
>
> **Q1)** If my understanding is correct, the paper assumes that the support of the test distribution is contained within that of the training distribution. This seems quite restrictive -- what if there are some unseen prompts or responses?
>
> ***A1)*** You are correct in your understanding: our paper assumes that the support of the test distribution is contained within that of the training distribution. Please note that the prompts and responses from both support sets are already different. We acknowledge that this is a restrictive assumption, particularly as it does not account for cases where the support of the test distribution is wider or only partially overlaps with the training distribution, which would indeed pose greater challenges. While we are considering the incorporation of these more complex assumptions in future work, we chose to limit the scope of our current work to address distribution shifts associated with the more manageable assumptions in LM alignment.
>
> ---
>
> **Q2)** In addition, what if the learner cannot obtain a representative validation dataset from the test distribution -- can anything still be done here?
>
> ***A2)*** In transfer learning, it is important to have a validation set that reflects the test distribution. While many existing methods rely on a large but unlabeled validation set, our approach requires only a small labeled validation set, which may be feasible to collect in practice. If this requirement is too strong, we may further relax it by incorporating weak supervision techniques, as in [Sugiyama et al. (2022)]. For example, instead of a fully labeled validation set, we may rely on partially labeled data, such as a small positive-and-unlabeled set, to reduce the burden of obtaining labeled validation samples from the test distribution.
>
> ---
>
> **Q3)** The experiments are in very controlled settings that involve simple mixtures of two distributions. From the results in Table 4, the improvement in win/match rates from using importance weighting seems relatively modest (3 - 8%) -- it is unclear how much benefits they bring in more complex real-world settings.
>
> ***A3)*** As mentioned earlier, we are indeed interested in exploring more complex settings in future work. However, we believe that even with simple mixtures of two distributions, our experiments sufficiently represent certain aspects of real-world settings. While the observed improvement in win/match rates may seem modest (3-8%), it demonstrates that DPO can be consistently enhanced through importance weighting.
>
> ---
>
> **Q4)** In addition, It is unclear what win/match rates should be considered as meaningful targets, as there is no baseline evaluation using, e.g., DPO on the test distribution.
>
> ***A4)*** Due to the limited amount of data available from the test distribution, we are unable to do DPO on the test distribution, i.e., Oracle. However, in the Helpful-Harmless LM scenario, our results below indicate that our methods can achieve performance that approaches that of DPO trained on the training dataset of the test distribution, which consists of approximately 9,000 examples. Please note that SFT for this DPO baseline was also trained on the same training dataset. With a dataset of around 15,000 examples, which matches the size used for the training set in the setting, we would expect a minimum win rate around 80%.
>
> | Method | Win rate |
> |:---:|:---:|
> | SFT (on test dist.) | 68.04 ± 5.36 |
> | DPO (on test dist.) | 74.18 ± 3.43 |
> | SFT | 56.40 ± 5.12 |
> | DPO | 68.71 ± 3.45 |
> | WPO | 70.26 ± 4.05 |
> | IW-DPO-L | 70.50 ± 3.46 |
> | IW-DPO-R | 72.28 ± 4.62 |
>
> ---
>
> **Q5)** It would also be beneficial to include comparisons with approaches such as [Zhou et al. (2024)] or other domain adaptation techniques that do not use importance weighting.
>
> ***A5)*** We have included WPO as described in [Zhou et al. (2024)] in the updated version (see Table 4). Our results indicate that while WPO outperformed the standard DPO, it did not surpass IW-DPO.
>
> ---
>
> **Q6)** The current setting assumes a single training distribution and a single test distribution, but in real-world applications, distributions may shift and evolve over time. It would be nice if the proposed approach can be extended to handle dynamic changes.
>
> ***A6)*** We appreciate your observation regarding the potential for distributions to shift and evolve over time. While our current approach  addresses static distribution changes, we understand the importance of dynamic changes. We are interested in exploring extensions to our method that would enable it to handle these dynamic changes in future work.
>
> ---
>
> References
> - Machine Learning from Weak Supervision: An Empirical Risk Minimization Approach, Sugiyama et al. (2022)
> - WPO: Enhancing RLHF with Weighted Preference Optimization, Zhou et al. (2024)

---

> ### Author Response · Authors · 2025-03-30
> **Reply (part 2)**
>
> **Q7)** Could you provide more details on the experiments? For example, what are the sizes of the datasets used for training and testing in each setting?
>
> ***A7)*** Yes, we have provided the exact number of examples for the training, validation, and test sets in Appendix B.2. This additional information clarifies the sizes of the datasets used for each setting.
>
> ---
>
> **Q8)** Why does the third setting use 45% test distribution data for training, whereas the first two only incorporate 25%?
>
> ***A8)*** We apologize for the confusion regarding the percentage of test distribution data used in the third setting. The correct value is 30%, not 45%. Although this is still lower than the percentages used in the first two settings, the decision to use 30% was made due to the limited size of the dataset. Further reducing the training percentage would have resulted in an even smaller training set.
>
> ---
>
> **Q9)** The paper is generally easy to follow. For clarity, in Section 3.2, it would be nice if you could briefly explain the motivation for applying a transformation function.
>
> ***A9)*** When working with complex data that requires deep models, estimating importance weights also requires powerful models capable of handling such data. One approach is to directly model the importance weights $w^*(x,y_1,y_2,b)$ using a deep neural network, which needs joint training of both an LM and a separate weighting model [Grangier et al. (2023)]. In contrast, we adopt a simpler approach that uses a transformation function derived from the LM to convert the inputs $(x,y_1,y_2,b)$ into the low-dimensional transformed data $z$ [Fang et al. (2020)]. This allows us to estimate the importance weights directly on the transformed data, without the need to train an additional deep model. To improve clarity, we have added such a brief explanation of the motivation for using a transformation function in Section 3.2.2.
>
> ---
>
> **Q10)** In Section 3.3.2, it would be helpful to refer to equation (5) when discussing reward values.
>
> ***A10)*** Yes, we have added a reference to equation (5).
>
> ---
>
> **Q11)** In Section 4.1.1, for evaluation metrics, it is not entirely clear what chosen responses and generated responses mean.
>
> ***A11)*** To clarify, in the test dataset, each instance consists of a prompt paired with one chosen response and one rejected response. The chosen response is treated as a reference response that represents the desired output for the test distribution. During the evaluation process, we provide the trained LM with a prompt and use it to generate a response (i.e., the generated response). We then compare this generated response to the associated reference response. We have added footnote 5 to elaborate on this.
>
> ---
>
> **Q12)** While related work is discussed in Section 1, a more thorough review would be beneficial, given the extensive research on importance weighting, domain adaptation, and distribution shift in both traditional settings and LLM domains. In addition, how is this problem related to pluralistic alignment [https://pluralistic-alignment.github.io/] -- are there techniques there that can be applied to handle distribution shift?
>
> ***A12)*** We have expanded our discussion in Section 2.1 and discussed the relation of our work to pluralistic alignment in Appendix D.
>
> ---
>
> References
> - Rethinking Importance Weighting for Deep Learning under Distribution Shift, Fang et al. (2020)
> - Adaptive Training Distributions with Scalable Online Bilevel Optimization, Grangier et al. (2023)

---

### Author Response · Authors · 2025-03-30
**Summary of the Revision**

We have improved our manuscript based on your feedback, addressing all your questions and incorporating the changes you requested. We have uploaded an updated version of the manuscript. The newly added text is highlighted in blue for clarity. However, we have not made the newly added tables and figures (captions) blue.

Some sections have been reorganized as follows:
- We have created subsections under Section 2. We have moved our discussions of importance weighting in language models from Section 1 to Section 2.1.
- We created subsections under Section 3.2. We discuss training objective derivation in Section 3.2.1 and weight estimation in Section 3.2.2.
- We have added subsections to Section 4.2 for better referencing.

Here is a brief summary of new experiments
- We have conducted experiments with larger models: Llama 3.1-8B-Instruct and Gemma 2-9B. Overall, the results are consistent with those from previous experiments involving smaller models. The results of these newly tested models are presented in Tables 4, 5, and 6, as well as Figure 4, while the results from the previous experiments with smaller models have been moved to Appendix C.
- We have included WPO [Zhou et al. (2024)] as another baseline method in Table 4. Overall, WPO did not surpass our method although it outperformed the standard DPO.
- We investigated the performance under various density ratio estimators as shown in Table 6. Our results indicate that the choice of density ratio estimators does not result in significant differences in performance.

We have also added grid lines to all figures in Section 4.

Thank you very much for your time and consideration.

---

References
- WPO: Enhancing RLHF with Weighted Preference Optimization, Zhou et al. (2024)

---

### Decision · Action_Editor_uWpc · 2025-07-01

**Recommendation:** Accept with minor revision

**Additional Comments:**

The paper studies the problem of mitigating deployment distribution shift in LLM alignment between training and test environments. The authors propose Importance-Weighted Direct Preference Optimization (IW-DPO), with the key idea of using a small amount of validation data from the test distribution to estimate weight and reweight training loss. The authors introduce two variants of their method and evaluate them across three semi-synthetic distribution shift environments. Empirical results showed that the proposed methods outperform DPO without importance weighting.

The reviewers all agree that the problem is practical and well motivated, the proposed solution is intuitive, and the writing is easy to follow. The reviewers raised several concerns, including
- Limitations in experiments such as missing baseline, relatively small models, and controlled synthetic environments. The authors addressed most of the concerns in revision with new experimental results on new baseline and larger models (Llama 3.1-8B-Instruct and Gemma 2-9B). New results still support the claims. However, the concern about relatively modest improvements in controlled synthetic environments still remains.
- Unclear discussion such as missing discussion with related works and theoretical assumption about test distribution. The authors provided refined related work discussions. The assumption that the support of the test distribution is within that of the training distribution is not uncommon but should be clarified in the main paper.

Overall, the paper presents a novel and intuitive solution for an important problem. While the paper could benefit from further discussing its limitations, the claims and evidence are sound and would be interesting to the audience working on LLM alignment. I highly suggest the authors make the following modification for camera-ready version:
- Please clearly discuss the assumption about the support of the test distribution in the main paper and its limitations.
- Further motivate the setting of the synthetic scenarios and acknowledge the modest or large improvments in different experiments.

**Audience:**

Yes

**Audience Explanation:**

Yes, the topic of mitigating distribution shift of RLHF method is interesting to audience.

**Claims And Evidence:**

Yes

**Claims Explanation:**

Yes. The claims are supported by the empirical results, which are greatly enhanced after revision.

---

> ### Author Response · Authors · 2025-07-23
> **Camera-Ready Submission**
>
> Dear AE,
>
> Thank you very much for your feedback and for overseeing the review process of our manuscript. We would also like to thank the reviewers for their reading and valuable feedback throughout the review process. All comments and suggestions have helped us improve the paper.
>
> We have carefully addressed all the points raised in the decision letter and the reviews, and we are pleased to submit the camera-ready version of the manuscript.
>
> Sincerely,
>
> Authors of 4220